Manuscript prepared for Ocean Sci.
with version 2015/09/17 7.94 Copernicus papers of the LATEX class copernicus.cls.
Date: 16 November 2016

# Seasonal resonance of diurnal coastal trapped waves in the southern Weddell Sea, Antarctica

Stefanie Semper[1] and Elin Darelius[1,2]

[1]Geophysical Institute, University of Bergen, and Bjerknes Centre for Climate Research, Bergen, Norway
[2]Uni Research Climate, Bergen, Norway

*Correspondence to:* stefanie.semper@uib.no

**Abstract.**

The summer enhancement of diurnal tidal currents at the shelf break in the southern Weddell Sea is studied using velocity measurements from 29 moorings during the period 1968 to 2014. Kinetic energy associated with diurnal tidal frequencies is largest at the shelf break and decreases rapidly

with distance from it. The diurnal tidal energy increases from austral winter to summer by, on average, 50 %. The austral summer enhancement is observed in all deployments. The observations are compared to results from an idealised numerical solution of the properties of coastal trapped waves (CTWs) for a given bathymetry, stratification and an along-slope current. The frequency at which the dispersion curve for mode 1 CTWs displays a maximum (i.e. where the group velocity is zero

and resonance is possible) is found within or near the diurnal frequency band, and it is sensitive to the stratification in the upper part of the water column and to the background current. The maximum of the dispersion curve is shifted towards higher frequencies, above the diurnal band, for weak stratification and a strong background current (i.e. austral winter-like conditions) and towards lower frequencies for strong upper layer stratification and a weak background current (austral summer).

The seasonal evolution of hydrography and currents in the region is inferred from available mooring data and conductivity-temperature-depth profiles. Near-resonance of diurnal tidal CTWs during austral summer can explain the observed seasonality in tidal currents.

## 1 Introduction

The shelf break region in the southern Weddell Sea (Fig. 1) is an area of great climatic interest.

This is where cold and dense water masses, formed on the continental shelf and underneath the Filchner-Ronne Ice Shelf (FRIS), cross the shelf break and descend the continental slope (Foster and Carmack, 1976; Foldvik et al., 2004; Nicholls et al., 2009), ultimately contributing to the formation of Antarctic Bottom Water which spreads out into the major oceans at abyssal depths (Orsi et al., 1999). Furthermore, warm off-shelf water, referred to as Warm Deep Water (WDW), crosses the

shelf break during austral summer in the form of modified Warm Deep Water (MWDW, Årthun

et al., 2012). The MWDW flows southward towards the Filchner Ice Shelf along the eastern flank of the Filchner Depression (Foldvik et al., 1985a, see map in Fig. 1 for location) and reaches, at least occasionally, the Filchner Ice Shelf front during austral autumn (Darelius et al., 2016). Some climate models suggest a larger inflow and a dramatic increase in basal melt rates below the FRIS within the next century (Hellmer et al., 2012).

Physical processes at the shelf break and on the continental slope influence both the cold outflow and the warm inflow in terms of their hydrographic properties and strengths. The variable depth of the thermocline, for example, which is controlled mainly by wind forcing and eddy overturning (Sverdrup, 1953; Nøst et al., 2011) will determine if and when warm water can access the continental shelf (Årthun et al., 2012). Meanwhile, variability in off-shelf water properties will alter the density contrast between the cold outflow and the ambient water, and thus the strength of the geostrophically balanced outflow (Kida, 2011; Wang et al., 2012). It will also influence the properties of the descending dense plume, since it is a mixture of outflow water and ambient water (Darelius et al., 2014). The co-location of the critical latitude for the tidal component $M_2$ and a critical slope leads to enhanced turbulence levels in the region (Fer et al., 2016). Mixing can be expected to be further enhanced at the shelf break by the strong diurnal tidal currents (Fer et al., 2015; Pereira et al., 2002). The strong diurnal tidal currents in the study region have been linked to the presence of continental shelf waves (Foldvik and Kvinge, 1974; Middleton et al., 1987; Foldvik et al., 1990), a class of coastal trapped waves (CTWs).

CTWs can be generated by e.g. tides (Thomson and Crawford, 1982) or wind (Huthnance, 1995). Additionally, a connection between the generation of the waves and the outflow of dense shelf water through troughs has been suggested (Marques et al., 2014; Jensen et al., 2013). CTWs with sub-inertial frequencies propagate along a trapping boundary, e.g. a coastal wall or a sloping bottom (Huthnance, 1995; Huthnance et al., 1986). The waves require the support of such a boundary to exist, and their energy decays exponentially with increasing distance from it (Mysak, 1980). While the direction of phase propagation is with shallow water to the left (in the southern hemisphere), the group velocity $c_g$ of CTWs, and thus the energy associated with the waves, can propagate in either direction (Fig. 2). If the group velocity is zero, i.e. for a maximum in the dispersion curve of a wave, energy cannot propagate. When the frequency of this maximum (hereafter called "resonant frequency", RF) in the dispersion relation coincides with the frequency of tidal forcing, resonance may occur and tidal currents will be amplified. In practice, energy likely escapes in one or the other direction along the slope. Leakage of energy occurs for example because of irregularities in the bathymetry and because the bottom slope changes (i.e. isobaths converge or diverge, Thomson and Crawford, 1982). Therefore, we use the term near-resonance rather than resonance.

Such near-resonant diurnal CTWs were first recorded on the shelf of the Outer Hebrides of Scotland by Cartwright (1969) and have been observed and modelled at numerous occasions and locations since then (e.g. Huthnance, 1974; Crawford and Thomson, 1982; Heath, 1983; Hunkins, 1986;

Padman et al., 1992; Skarðhamar et al., 2015). In our study region, Foldvik and Kvinge (1974) and Foldvik et al. (1985b) first suggested that CTWs caused the observed strong diurnal tidal currents, and they attributed a weakening of currents in austral winter to a seasonally varying stratification. Later, Middleton et al. (1987) and Foldvik et al. (1990) found a particularly strong enhancement of the $K_1$ tidal constituent during austral summer. The summer maximum was hypothesised to be due to the interaction of barotropic CTWs with topography in the presence of a seasonally variable mean current (Foldvik et al., 1990). The authors showed how changes in the background current will affect the phase of diurnal CTWs, which are assumed to be generated upstream, as they arrive in the study region. These studies were based on a small number of moorings and a barotropic shelf wave model neglecting the effects of stratification. Studies by e.g. Marques et al. (2014); Jensen et al. (2013); Brink (1991); Wang and Mooers (1976) indicate that, in addition to the currents, seasonally varying hydrography alters the properties of CTWs.

Another seasonal phenomenon which can potentially influence the CTW generation is sea ice. Frictional damping of tidal CTWs due to sea ice is suggested to be the cause of the observed reduction of tidal currents over the shelf in the Sea of Okhotsk during winter when the sea ice cover exceeds 80% (Ono et al., 2008). Strong tidal currents in turn are important for local sea ice deformation (Padman et al., 1992) and hence sea ice concentration (Mack et al., 2013). A study by Nakayama et al. (2012) showed that CTWs can locally enhance sea ice drift.

To the east of our study region, the continental slope steepens considerably (Fig. 1), and isobaths hence diverge in the direction of CTW propagation. Numerical simulations from the Barents Sea region showed that tidally generated CTWs were confined to a region of divergent bathymetry (Skarðhamar et al., 2015).

Our study expands on previous investigations of tidally generated CTWs at diurnal frequencies in the shelf break region of the southern Weddell Sea. Records from 29 moorings collected over more than four decades are used to quantify the strength of the diurnal tidal currents in the area and to describe their temporal and spatial variability. The extended data set confirms the existence of an enhancement of the diurnal tidal currents during austral summer and shows that it is a persistent phenomenon. Previous studies – based on a small sub-set of our data records – have suggested that the summertime enhancement is due to changes in the oceanographic "background", as it determines the dispersion relation for the CTWs which are responsible for the tidal amplification in the area. We investigate this further by using a numerical code (Brink, 2006) to study the sensitivities of the CTW properties and the RF to seasonal changes in hydrography and background current. With this aim, we synthesise observational data from the continental slope providing a novel description of the seasonal changes in shelf break hydrography. The results from the numerical code suggest that the effect of a changing stratification dominates the effect of the background current. Finally, we investigate the role of divergent bathymetry and the seasonal variability of sea ice cover in the southern Weddell Sea, and we discuss the potential relevance for tidally generated CTWs.

## 2 Data and methods

Current meter data from 29 moorings (Foldvik et al., 2004; Jensen et al., 2013; Darelius et al., 2016) located on the continental slope and shelf in the area surrounding the Filchner Depression have been analysed. The records span the years 1968 to 2014 and are each of 1–2 years duration. The locations of the moorings are shown in Fig. 1, and deployment details are listed in Table 1. The mean currents included in Fig 1 are vertical means for moorings M1 to M5, while for moorings C, W2, W3, F3 and F4 only the uppermost instrument has been considered to minimise the effects of the Ice Shelf Water plume. Temperature records suggest that plume water is rarely present at these levels.

The coordinate system is rotated clockwise to align the $y$-axis with the isobaths, agreeing with the set-up of the numerical code (Brink, 2006) in the southern hemisphere. $u$ is thus directed on-shelf and $v$ along the continental slope (Fig. 1). The rotation angle $\varphi$, positive for clockwise rotation, is listed in Table 1; it is inferred for each mooring from the local bathymetry based on the GEBCO_2014 bathymetry grid (The GEBCO_2014 Grid, version 20150318, http://www.gebco.net) and using an average length scale of the order of 10 km. The estimated accuracy of $\varphi$ is approximately $\pm 10°$.

Time series of kinetic energy (KE) associated with the diurnal tidal currents are constructed as follows: The hourly-averaged current meter data are divided into intervals of 1.5 months length beginning every 14th day. For each interval, the power spectral densities are estimated using Welch's method (Welch, 1976) and 14 day long, 50 %-overlapping Hanning windows. The diurnal tidal KE is obtained by integrating the velocity spectra,

$$\text{KE} = \int\limits_{\omega_1}^{\omega_2} (S_u + S_v)\, d\omega, \tag{1}$$

where, following Jensen et al. (2013), $\omega_1$ and $\omega_2$ correspond to periods of 26.9 h and 21.3 h respectively.

Diurnal tidal KE has also been inferred using tidal predictions from the Circum-Antarctic Tidal Simulation version 2008b (CATS2008b), an updated version of the linear tidal inverse model described by Padman et al. (2002). The barotropic currents at the specific tidal frequencies are predicted for the respective time and location of every mooring deployment, and tidal KE is directly calculated as monthly running mean from the amplitudes of the predicted tidal currents.

Tidal ellipses, i.e. major and minor axes, inclinations and Greenwich phases, have been obtained from the mooring records using harmonic analysis (T_TIDE, Pawlowicz et al., 2002), a Matlab version of the FORTRAN code developed by Foreman (1978).

Records of temperature and salinity from mooring M3, located at the 725 m isobath just east of the Filchner Depression sill (Fig. 1), are used to describe the seasonal changes in hydrography at the shelf break and upper continental slope. The mooring records are complemented by a conductivity-temperature-depth (CTD) profile obtained during the deployment cruise in 2009 and by hydrographic measurements obtained in the vicinity of the M3 location (within 10 km, Fig. 1) provided by seals

tagged with small CTD sensors (described in Årthun et al., 2012, hereafter referred to as "seal data"). The accuracies of the seals' temperature and salinity measurements are stated to be $0.005°$ C and 0.02, respectively (Boehme et al., 2009).

In addition, we use wind observations from Halley Research Station, located at 75° 35' S, 26° 39' W (Fig. 1), from 1957 to 2014 (British Antarctic Survey, 2013, updated 2014) and satellite derived records of sea ice concentration (Meier et al., 2013, updated 2015), available for the period 1978 to 2014. The sea ice concentration is averaged over the study area (inset in Fig. 1).

## 3   Observational results

### 3.1   Spatial and temporal variability of tidal currents

The diurnal tidal frequency band shows enhanced variance for both the $u$- and $v$-component, especially at the frequencies of the most important diurnal tidal constituents $K_1$ and $O_1$ (Fig. 3). High energy levels are additionally observed at semi-diurnal frequencies and around 35 h, as also found by Jensen et al. (2013) and Darelius et al. (2009).

The energy associated with the diurnal tidal currents, the diurnal tidal KE (Sect. 2), shows little variation with depth, except at the lowest measurement level at 25 m.a.b. In this bottom boundary layer, the diurnal tidal KE is slightly decreased compared to the overlying water column (Fig. 4). Depth-averaged diurnal tidal KE is used for further analysis.

Figure 5 shows the spatial distribution of diurnal tidal KE during austral summer. The magnitude of diurnal tidal KE is highest directly at the shelf break (e.g. moorings B2, F1, M3) and decreases rapidly with distance from it. The tidal currents rotate clockwise on the deeper continental slope and anticlockwise at the shelf break and on the shelf. The major axes of the tidal ellipses at the $K_1$ frequency are directed across the continental slope for moorings located at the shelf break and on the continental slope (especially for the ones east of the Filchner Depression and west of the ridge). Hence, the diurnal tidal energy is higher in the across-slope component than in the along-slope component. Tidal currents recorded at moorings on the shelf are close to circular.

Time series of diurnal tidal KE (Fig. 6a) show two local maxima; one in austral summer and one in austral winter. The austral summer maximum is 30 % to 180 % higher than the winter maximum. This amplitude difference is especially strong in records from moorings on the continental slope and at the shelf break, but it is observed in all deployments of sufficient length. For moorings on the continental shelf, the difference between the maxima is sometimes less pronounced (e.g. Fr2 in Fig. 6a). Tidal KE inferred from the tidal model CATS (Fig. 6b) shows two annual peaks of similar amplitude. The two peaks are the result of interference between the diurnal constituents (see Sect. 5), but astronomical forcing cannot explain the observed difference in tidal KE between austral summer and winter.

Estimates of wavelengths were obtained from mooring pairs M1–M4 and M2–M5, based on the difference in Greenwich phase obtained from T_TIDE. The mooring pairs were deployed roughly along the 1100 m and 1950 m isobaths at a separation of 71 km and 86 km, respectively. For the $O_1$ tidal constituent, the uncertainty and hence the range is large during austral summer (200–1600 km), while austral winter values are found in the range 300–600 km. The wavelength obtained for $K_1$ is in the range 250–500 km for all seasons.

## 3.2   Seasonal variability of the hydrography and current on the upper slope

The seasonal variability in the hydrography at the shelf break and on the upper slope is investigated by merging all available observational data (moorings, CTD, seal data) near the location of mooring M3 (Fig. 7a,b).

Cold and fresh Winter Water (WW) is found above warm ($0°$ C $< \theta < 0.8°$ C, e.g. Gammelsrød et al., 1994) and saline ($34.64 < $ S $< 34.72$) WDW, the Weddell Sea version of the Circumpolar Deep Water which composes most part of the Antarctic Circumpolar Current and which enters the Weddell gyre along its eastern rim (Ryan et al., 2016). The MWDW ($-1.7°$ C $< \theta < 0°$ C) that is able to intrude on the continental shelf is a mixture of WW and WDW. While the temperature in the upper approximately 400 m is near the freezing point year-round, the salinity of the surface layer increases from 34.0 in February to 34.4 in October. The cold and fresh surface layer during austral summer likely results from local sea ice melt.

The thermocline is found at a depth of approximately 400 m from December to April and deepens by 200 m to approximately 600 m during May to August. The deeper moorings (e.g. M5) show that seasonal changes in the water column below the thermocline are negligible (not shown). Generally, the seal data show higher salinities and temperatures at depth compared to the mooring data (also compared to the range of the unfiltered mooring records, not shown), suggesting that the WDW and the thermocline are found higher up in the water column in 2011 compared to 2009.

The density profiles (Fig. 7c) show a gradual increase in density at the surface from $\sigma_0 \approx 27.3$ kg m$^{-3}$ to 27.7 kg m$^{-3}$, indicating a relatively stable stratification in the upper part of the water column during austral summer and a relatively homogeneous, weakly stratified upper layer during austral winter.

Observations of the Antarctic slope current flowing westward along the shelf break in the Weddell Sea are relatively scarce, and our knowledge of its strength, width and variability in our study region are limited. Upstream, at $12°$ W, Fahrbach et al. (1992) observed a south-westward flowing current following the continental shelf break with annual mean velocities of 10 cm s$^{-1}$ to 20 cm s$^{-1}$ and a maximum velocity (hourly average) of over 60 cm s$^{-1}$. Although inconclusive, the records suggest a wind-driven seasonal cycle with a magnitude of about 5 cm s$^{-1}$ where maximum currents are observed in late austral autumn.

At $17°$ W, the core of the slope current is found above the 1000 m isobath with a westward surface velocity of 50 cm s$^{-1}$ (Heywood et al., 1998). The current is suggested to weaken towards Halley

Bay (Fahrbach et al., 1992), and at 27° W, it splits into two branches, where one branch follows the coast southwards and the other one continues along the continental slope (Gill, 1973) into our study region.

Mooring records from the region west of the Filchner Depression cover mainly the lower part of the water column, and the majority of the observations are greatly influenced by the Filchner
overflow plume (Foldvik et al., 2004), thus giving little information about the slope current. Fig. 1 shows annual mean currents from moorings and instrument levels in the area, that, based on the accompanying temperature records, are not directly affected by the dense outflow of Ice Shelf Water (Sect. 2).

East of the depression, the strongest along-slope currents are observed at mooring M3, relatively
close to the shelf break at the 750 m isobath. Here, the magnitude of the annual mean current is $10\,\mathrm{cm\,s^{-1}}$ to $17\,\mathrm{cm\,s^{-1}}$ (directed westward and stronger towards the bottom) while monthly mean values reach $25\,\mathrm{cm\,s^{-1}}$ during austral winter.

At mooring M4 (located at the 1050 m isobath, less than 10 km north of M3, Fig. 1), a much weaker westward ($3\,\mathrm{cm\,s^{-1}}$) mean current was observed, and austral winter values reached $8\,\mathrm{cm\,s^{-1}}$.
At M5 (1976 m depth, about 40 km north of M3), the magnitude of the annual mean current is $<1\,\mathrm{cm\,s^{-1}}$ with a peak in early austral winter of $2$–$3\,\mathrm{cm\,s^{-1}}$.

The limited observations available from our study region suggest a westward flowing jet, which is relatively narrow and appears to be centred at the shelf break. The jet intensifies and widens during early austral winter. Winter time intensification of the slope current is also observed by Núñez-Riboni
and Fahrbach (2009) and Graham et al. (2013).

## 4 Numerical code

### 4.1 Set-up

The numerical code described in Brink (2006) and adapted for the southern hemisphere by Jensen et al. (2013), is used to calculate the properties of stable, inviscid CTWs for different stratification,
bathymetry and mean flow.

The code was set up using 30 vertical levels and 120 horizontal grid points to represent a 2 D-cross-slope section. This is within the recommended range of grid points; increasing the resolution leads to instability and failure of the test for hydrostatic consistency. Following Jensen et al. (2013), we use a closed coastal but open offshore boundary, a free surface and a negligible bottom friction.
Furthermore, we apply the same bathymetry as Jensen et al. (2013). It represents an average of six across-slope sections with approximately 20 km separation in the area of moorings M1 to M5 and compares well to sections farther west in our study area (not shown).

The input stratification vector (squared buoyancy frequency, $N^2$) is linearly interpolated onto the vertical levels of the code and duplicated for the horizontal cross-shelf section before it is converted

to density, hence no across-shelf stratification changes are taken into account. If an along-shore current is specified, the background density field is altered by applying the thermal wind equation. The stratification at each level $n$ is then determined from the density difference between levels $n-1$ and $n+1$.

## 4.2 Sensitivity to stratification

A reference stratification profile was constructed based on all available CTD data collected in January and February in the eastern part of the study area. Similar profiles were constructed for areas farther to the west. Figure 8a shows the obtained density and stratification profiles, representative for the shelf break at moorings M1 to M5 in austral summer. A simplified version of the stratification profile (Fig. 8b) indicates the parameters changed in the sensitivity test: the strengths of the surface

magnitude (SM) and the subsurface magnitude (SSM) around 500 m depth, the depth of the SSM (SSD) and the constant magnitude at depths below 1200 m ("deep magnitude", DM). The values of the applied parameter values are listed in Table 2.

The dispersion curves and their group velocities for wave modes 1 to 3 corresponding to the reference stratification (Fig. 8a) are presented in Fig. 9. Mode 1 is the only wave mode for which

the dispersion curve shows a maximum, i.e. where the group velocity becomes zero. These results suggest that CTWs with a wavelength of approximately 1260 km and a period of approximately 30 h will be trapped while CTWs with tidal frequencies cannot exist. For tidal CTWs to exist, the dispersion curve must pass through the tidal band, i.e. the maximum of the dispersion curve (the resonant frequency, or "RF") must lie within (thus giving near-resonance) or above the diurnal tidal

frequency band.

As the numerical code has a vertical resolution of 160 m, defining the uppermost $N^2$ value, which ought to represent the considerable changes close to the surface, is not a straightforward task. For the reference stratification profile (Fig. 8), the surface $N^2$ value used in the numerical code is from 20 m depth. Using the surface profile value ("surface top" in Fig. 9) or an average of the upper

80 m ("surface mean" in Fig. 9) shifts the dispersion curve and thus the RF to higher frequencies (Fig. 9). For stratification profiles which are representative for areas farther west at the shelf break and constructed similarly to the reference stratification with surface $N^2$ values of the upper 80 m average ("stratification 2–4"), the dispersion curve and RF are similarly shifted to higher frequencies (Fig. 9).

Keeping in mind the variations along the shelf break and with different approaches on how to choose the uppermost stratification value, the characteristic parameters of the reference profile (SM, SSM, SSD, DM, Fig. 8b) are varied in the following to explore the general effects of stratification on the dispersion curve and the RF.

Figure 10 shows the results from the sensitivity test for stratification, where the RF is identified

from each dispersion curve obtained from the modified stratification input. An increase of $N^2$ at the

surface (case SM) leads to a decrease in RF, which moves through the diurnal tidal frequency band for the modelled range of surface stratification. Contrarily to case SM, an increase of the stratification maximum at approximately 640 m depth (case SSM) increases the RF. The effect of an increase in depth of the subsurface maximum (case SSD) results in an apparent decrease of the RF. However, due to the interpolation in the numerical code, the stratification around the subsurface maximum as well as the exact value of the maximum are difficult to preserve. Hence, the actual effect of case SSD appears to be rather small. Varying the stratification below 1200 m depth (case DM) has a negligible effect on the RF.

### 4.3 Sensitivity to along-slope current

The optional along-shore current in the numerical code has a Gaussian shape; its offshore, onshore, upward and downward $e$-folding length scales must be specified, in addition to the centre position, strength and depth of the current.

For the sensitivity test, a barotropic (i.e. with a large vertical length scale) westward current is assumed which is centred at the shelf break. The density is set to be undisturbed at the coast when the density field is altered according to the thermal wind equation, with the input $N^2$ vector being the reference stratification for all runs. The width and strength of the current are varied from 10 km to 100 km and magnitudes of $0.1\,\mathrm{m\,s^{-1}}$ to $0.5\,\mathrm{m\,s^{-1}}$, respectively (Fig. 11). This roughly encompasses the observed structure and variability of the slope current described in Sect. 3. Generally, both a stronger and a wider current lead to an increase in RF; with the effect of increased current strength being largest.

In another test, the location of the current core was moved 40 km on and off shore relative to the shelf break. The sensitivity of the RF decreases slightly when the current core is located off shore from the shelf. The magnitude of the change in RF for a 40 km off-shore shift depends on the width of the current, but it is comparable to a change in current velocity of $\pm 10\,\mathrm{cm\,s^{-1}}$ (Fig. 11).

Although the overall effect of an added barotropic slope current is minor compared to the sensitivity to changes in stratification (cf. $y$-axes in Fig. 10 and Fig. 11), the sensitivity depends noticeably on the vertical length scale. As an example, a 40 km wide current with a westward core velocity of $0.2\,\mathrm{m\,s^{-1}}$ and a reduced downward $e$-folding length scale (2000 m instead of 4300 m) is chosen. The RF is then considerably larger (open circle in Fig. 11) than for the more barotropic case.

## 5   Discussion

Observations from the continental slope in the southern Weddell Sea show anomalously strong tidal currents at diurnal frequencies (Middleton et al., 1987). Our extended analysis – including all current meter records (1968–2014) from the region – confirms previous findings suggesting that the strong currents are the result of tidally forced CTWs (Middleton et al., 1987; Foldvik et al., 1990, 1985b).

The observations agree qualitatively with the mode 1 CTW generated in the numerical code pro-
vided by Brink (2006). As expected, the rotational direction of the observed and simulated currents
changes from anticlockwise on the upper part of the continental slope to clockwise on the deeper part
of the slope, and the strength of the diurnal tidal currents increases towards the shelf break (Fig. 12).
The wavelengths inferred from the observations are generally consistent with those obtained from
the numerical code for cases where CTWs of diurnal frequencies are allowed.

Time series of the KE associated with the diurnal tides show a persistent pattern with two annual
peaks, one in austral summer and one in austral winter, and minima in spring and autumn. The austral
summer peak is enhanced for all moorings by 30 % to 180 % compared to the austral winter peak.

The semi-annual signal, i.e. the two peaks, is at least partly the result of astronomical forcing:
around the equinoxes the sun is nearly above the equator (and also the moon within about a fortnight
so that both their declinations are small); hence the diurnal forcing is at its minimum. In June and
December, the solar declination is at its maximum and the solar contribution to the diurnal tide is
greatest.

This results in two semi-annual peaks of diurnal tidal KE with similar amplitude (Fig. 6b). At the
mooring locations M3 and M5, the diurnal KE of the tides directly calculated from the prediction
of the CATS tidal model is reduced by about 20–25 % near the equinoxes, while the observations
typically show a reduction of more than 50 % compared to the winter values.

It is possible that other factors contribute to the observed semi-annual variability. For example,
there may be a semi-annual cycle of mixing (and hence stratification) caused by the semi-annual
tidal signal. Alternatively, the observed variability of diurnal KE may be the sum of the effect of
an annual cycle in stratification that is phase-shifted relative to the effect of changing background
current (L. Padman, personal communication, 9 August 2016).

We hypothesise that the enhancement of tidal KE during austral summer is caused by near-
resonance of diurnal CTWs. The obtained dispersion relations suggests that diurnal CTWs may be
near-resonant, i.e. that the group velocity is zero or close to zero at diurnal frequencies so that energy
cannot propagate out of the area, which results in amplified diurnal tidal currents. Near-resonance of
diurnal CTWs in the study region was also suggested by Middleton et al. (1987) using a barotropic
shelf wave model (Saint-Guily, 1976). The dispersion relation is shown to be relatively sensitive to
changes in the upper ocean stratification, and the observed seasonal changes in upper ocean hydrog-
raphy discussed below causes the RF to move through the diurnal tidal band, so that diurnal CTWs
are "nearer" resonance during austral summer than during austral winter, when the RF falls above
the diurnal band. Tidal currents may be enhanced for a range of frequencies surrounding the RF,
and thus the RF does not need to coincide with one exact tidal frequency for amplification to occur
(Chapman, 1989).

The largest seasonal changes in the shelf break hydrography in the region occur above the pyc-
nocline, similar to regions farther east in the Weddell Sea (Nøst et al., 2011; Graham et al., 2013).

Cooling and a gradual increase in salinity (due to ice freezing and brine rejection) during austral autumn and winter leads to a gradual deepening of the surface layer. Towards the end of the winter (August–September) the upper 400 m are relatively homogeneous. During austral summer, the winter layer is capped by a fresh and relatively warm surface layer which likely is the result of local sea-ice melt and solar heating. The layer of summer surface water is thin (10–100 m, see CTD-profile in Fig. 7) and greatly increases the stratification by creating a seasonal, shallow pycnocline. The sensitivity test (Fig. 10) shows that the value of the RF is sensitive to the stratification in the upper layer (SM) and that it increases for decreasing stratification. The response in RF to realistic changes in SM is of sufficient magnitude to cause the RF to move through the diurnal tidal band.

While the RF is influenced by changes in the strength of the permanent (deeper) pycnocline (SSM), which is the transition from WW above to MWDW and WDW below, there is no observational evidence suggesting that it would change in magnitude. The depth of the permanent pycnocline (SSD), however, increases from about 400 m in austral summer to about 600 m, but changes in SSD have little or no influence on the RF.

Despite the fact that the stratification affects the dispersion curve of CTWs considerably (Fig. 9), CTWs are relatively barotropic in the region (Fig. 4 and Fig. 12; Middleton et al., 1987; Jensen et al., 2013). This is expected since the Burger number is $Bu = \left(\frac{NH}{fL}\right)^2 \approx \mathcal{O}(10^{-4}) \ll 1$, where $N^2$ is the stratification ($N = (0.97–1.7)\cdot10^{-3}\,\mathrm{s}^{-1}$), $f$ the Coriolis factor ($-1.4\cdot10^{-4}\,\mathrm{s}^{-1}$) and where $H$ (400–600 m) and $L$ (300–500 km) are representative depth and length scales, respectively.

Foldvik et al. (1990) suggested that the observed seasonality was linked to the variability of the slope current. The available observations of the slope current are limited and do not allow a detailed description, but the data from moorings M1 to M5 suggest, in agreement with observations upstream (Nøst et al., 2011; Graham et al., 2013; Núñez-Riboni and Fahrbach, 2009), that the westward flowing slope current is intensified during austral winter. When a barotropic, westward background current is included in our numerical set-up, the dispersion curve (and thus the RF) is shifted toward higher frequencies (Fig. 11 and Jensen et al., 2013), but the effect is small compared to the effect of stratification changes. The combined effect of stratification and current, however, is considerable, as the stronger current observed during austral autumn and winter will add to the effect of the low winter time stratification and move the RF to higher frequencies.

The low values of tidal KE during austral winter can potentially be caused by frictional damping of CTWs by sea ice. When sea ice is in free drift (Padman and Kottmeier, 2000), no tidal energy is dissipated at the ocean-ice interface. However, as ice concentration increases and internal ice stresses prevent the ice from responding to local tidal currents, the stress at the ocean-ice interface may be significant compared with friction at the seabed, thus removing tidal energy and reducing tidal currents (Padman et al., 1992). This mechanism was suggested by Ono et al. (2008) to explain the reduction in tidal CTW energy observed at at one of their mooring sites during winter. The reduction was only observed at the site away from the region of CTW generation, indicating the

cumulative effect of frictional dampening over distance. However, the authors fail to explain why the period with reduced tides are much shorter (and misaligned) compared to the period with dense (>80 %) sea ice cover.

The ice cover in the study region normally exceeds 90 % during austral winter (mid-April to mid-November) and decreases to a minimum of on average 50 % in February (Fig. 14), and high (low) sea ice concentration hence coincides with low (high) diurnal tidal energy levels, as expected if frictional damping by sea ice is important. The semi-diurnal tidal currents, however, are observed to be larger during austral winter than during summer (Foldvik et al., 1990), which is inconsistent with the response if damping by ocean-ice interactions was a significant factor. We can unfortunately neither quantify nor rule out the relevance of sea ice concentration and sea ice damping to the observed seasonality of the diurnal tidal currents.

While the tidal force is the main generation mechanism for CTWs in the diurnal tidal band (Thomson and Crawford, 1982), CTWs can also be generated by wind (Huthnance et al., 1986). Short duration storms have been observed to excite near-resonant CTWs of mode 1 (Gordon and Huthnance, 1987), i.e. the response to storms would in our case resemble the tidally forced waves. Time series of wind from the nearby Halley Research Station (see Fig. 1 for location), however, show that storms (wind speed $>20 \, \mathrm{m \, s^{-1}}$) are rare during austral summer and, as expected, more frequent during winter and early spring. CTWs induced by storms can hence not explain the summer enhancement of the diurnal tidal currents. Fourier analysis of the time series reveals a daily cycle in wind strength with an increase of magnitude of up to $1.4 \, \mathrm{m \, s^{-1}}$ around noon, which likely results from local boundary layer effects: the stable boundary layer which develops during the night is destroyed during the day by mixing due to solar insolation (see, e.g. Stull, 2012). Since the signal is weak, we conclude that these oscillations are not responsible for the observed summer amplification.

The CTWs owe their existence to topography, and the dispersion curve is sensitive to changes in bathymetry (e.g. Jensen et al., 2013, and Fig. 9). Just east of our study region, the continental slope is much steeper (Fig. 1), i.e. the isobaths diverge towards the west. Divergent bathymetry has been shown to favour the generation of diurnal CTWs (Skarðhamar et al., 2015). On a divergent slope, tidal energy travelling along the slope may slow down and converge as the group velocity decreases due to the changing topography, while energy travelling in the other direction will speed up and diverge. There are no direct observations of tidal currents from the steep, eastern region, but the tidal motion of sea ice above the shelf break there suggests weaker currents compared to the study region (Padman and Kottmeier, 2000). The wavelengths of the diurnal CTWs are typically large compared to the length scale over which the changes in bathymetry discussed above occur and the scale of other topographic features in the area. While the implications of this are beyond the scope of the current study, we note that the CTWs modelled by Skarðhamar et al. (2015) similarly have wavelengths which are considerably larger than the topographic scales. A full 3 D-analysis, similar

to the one by Skarðhamar et al. (2015), would be needed to fully explore the effect of bathymetry on CTWs in the study region.

The anomalously large diurnal tidal currents and the CTWs will potentially influence the exchange of MWDW across the shelf break. In an idealised model study, CTWs were shown to enhance the inflow of warm water through a trough cross-cutting the continental shelf (St-Laurent et al., 2013), similar to the Filchner Depression. We note that the depth of the WW–WDW transition (here identified by the -1°C isotherm) varies on diurnal time scales (Fig. 13a), and e.g. in December 2009, the vertical excursion of the isotherm associated with the diurnal tides is >100 m (Fig. 13b). The depth of the transition is likewise affected by CTWs with 35 h period (Fig. 13; Jensen et al., 2013). The existence and strength of diurnal (and longer period, Jensen et al., 2013) CTWs in the region must hence be expected to directly influence the availability of warm water above the shelf depth, i.e. at depths where it can potentially access the continental shelf through the influence of other processes such as e.g. a background mean flow, rectified tidal flows, friction or eddy exchanges. In addition, the tides in the area will greatly influence mixing (Fer et al., 2016) and hence modify the stratification in the shelf break area. Modelling efforts aiming to describe and predict the oceanic heat transport towards the FRIS cavity thus ought to include tidal forcing to correctly capture the dynamics at the shelf break.

Finally, we mention that the observed diurnal tidal currents are up to one order of magnitude larger than those predicted by the tidal model CATS2008b (Fig. 12, Padman et al., 2002), which does not include stratification or the variability of mean circulation required to predict seasonal modulation of tidal currents. Moreover, due to discrepancies in the model bathymetry, the predicted peak tidal currents are not consistently aligned with the shelf break when running CATS along a cross-shelf section through the locations of moorings M1 and M2 (see Fig. 12 and Fig. 1 for location of section). Hence, care must be taken when using CATS to de-tide velocity observations from the study region.

## 6 Conclusions

Velocity measurements at 29 moorings located on the continental slope and shelf in the southern Weddell Sea from the period 1968 to 2014 show pronounced diurnal tidal variability. Diurnal tidal currents are strongest at the shelf break and substantially enhanced during austral summer. The summer enhancement is not predicted by the tidal model CATS2008b (Padman et al., 2002), as the model does not include stratification or the variability of mean circulation. We investigated the possibility for near-resonant CTWs causing the enhanced diurnal tidal currents by using a 2 D-numerical code to obtain CTW properties (Brink, 2006). Dispersion curves of mode 1 CTWs have a maximum in frequency (the resonant frequency, or "RF"), which results in zero group velocity, i.e. trapped energy. The RF moves in and out of the diurnal tidal frequency band depending on the stratification and the slope current which both vary seasonally as hydrographic and current observations at the shelf break

455 reveal. For the weakly stratified water column and strong slope current during austral winter, the RF is found above the diurnal band, suggesting the generation of weak, non-resonant tidal CTWs which quickly propagate out of the generation area. For austral summer conditions, i.e. a more stratified upper water column combined with a weaker slope current, the RF can fall into the diurnal band, thus leading to near-resonant diurnal CTWs enhancing the tidal currents.

460    While no direct influence of wind on the diurnal tidal currents and no evidence of sea ice affecting the diurnal CTWs have been found, the varying bathymetry east of the study area likely affects the propagation of the CTWs. Studies with realistic 3 D-ocean models are needed to quantify these influences as well as to detect the generation site of the CTWs.

The shelf break region in the southern Weddell Sea is an area of great climatic interest. Cold, dense
465 water descends the continental slope and contributes eventually to the formation of Antarctic Bottom Water, while warm WDW and MWDW flowing onto the shelf prospectively may reach the cavity below FRIS, thus enhancing basal melt rates. The strong diurnal tidal currents at the shelf break facilitate the cross-shelf exchange of water masses and contribute to mixing, hence influencing the hydrographic properties of both the cold outflow and warm inflow.

470    *Acknowledgements.* For deployment and recovery of moorings, we would like to thank AWI and the crew and scientists on RV *Polarstern* cruises PS08 (recovery of moorings D1, D2, S2-1985 and S3), PS12 (deployment S2-1987), PS34 (deployment Fr1 and Fr2), PS53 (recovery F1–4) and PS82 (recovery SB, SC, SD and SE). Thanks to K. Brink for sharing the numerical code and I. Fer for helpful comments and suggestions. We would also like to thank two anonymous reviewers and L. Padman as well as the editor J. Huthnance for constructive
475 comments which significantly improved the manuscript. The research was partially funded by the Centre for Climate Dynamics at the Bjerknes Centre and by the NFR funded project WARM (231549).

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

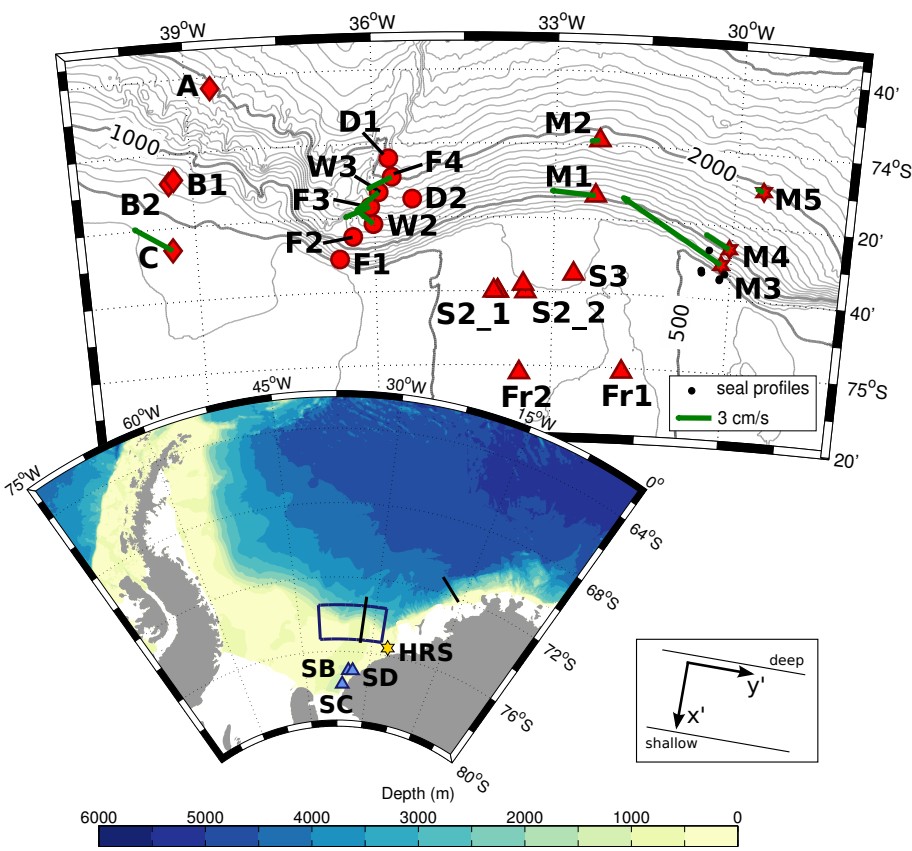

**Figure 1.** Map of the Weddell Sea (bathymetry from The GEBCO_2014 Grid, version 20150318, http://www.gebco.net) and locations of moorings in the study area (shapes according to their location west, along the ridge, at and in the Filchner Depression, or east on the continental slope). S2_1 and S2_2 show the location of mooring S2 prior to and after 2000. Annual mean currents (see Sect. 2) are indicated by green arrows. The study area is also the area the sea ice concentration has been averaged over. The yellow star marks Halley Research Station (HRS), while the two black lines indicate cross-slope sections used for the bathymetry test in the numerical code and to derive diurnal tidal KE from the tidal model CATS. The inset in the lower right corner illustrates the orientation of the coordinate system.

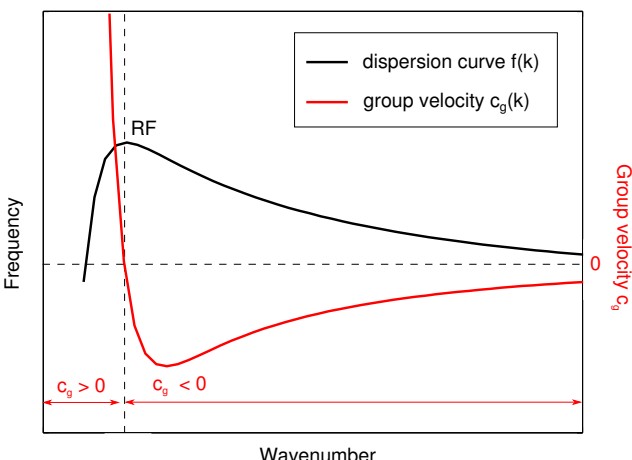

**Figure 2.** Illustration of a typical mode 1 CTW dispersion curve (black line) with the corresponding group velocity (red line). For small wavenumbers, the group velocity is positive, while it becomes negative for larger wavenumbers, i.e. indicating a change of direction of energy propagation. At the maximum of the dispersion curve (the resonant frequency, or "RF"), the group velocity vanishes and energy is trapped.

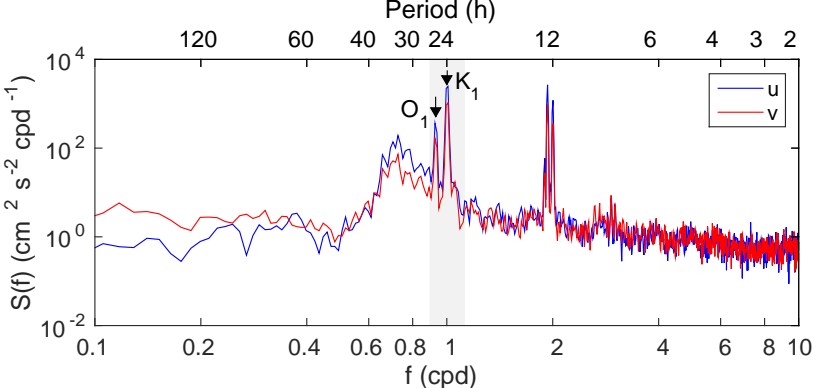

**Figure 3.** Power spectral density of depth-averaged, rotated $u$- and $v$-velocity components at mooring M3. The diurnal tidal frequency band around one cycle per day (cpd) is marked in grey; black arrows indicate frequencies of the $K_1$ and $O_1$ tidal constituents.

**Table 1.** Name, recording year, location in degrees and minutes, bottom depth, number of recording days, depths of current meters or ADCPs (in format "first level:depth increment:last level") in metres above bottom (m.a.b.) and angle $\varphi$ for the clockwise rotation of the coordinate system for all moorings used in this study. For some moorings located on the flat shelf or close to varying bathymetry (i.e. at the ridge), the rotation angles have increased uncertainty (marked with an asterisk). More details on the moorings can be found in Foldvik et al. (2004) and references therein, as well as in Jensen et al. (2013) and Darelius et al. (2016).

| Mooring | Year | Latitude | Longitude | Depth (m) | Recording days | Instrument depth (m.a.b.) | $\varphi$ |
|---|---|---|---|---|---|---|---|
| B1 | 1968 | -74 07 | -39 18 | 657 | 265 | 23 | 122 |
| B2 | 1968 | -74 08 | -39 23 | 663 | 460 | 23 | 122 |
| S2-1977 | 1977 | -74 40 | -33 56 | 558 | 411, 257 | 25, 100 | -10* |
| C | 1977 | -74 26 | -39 24 | 475 | 630, 631 | 25, 125 | 127 |
| A | 1978 | -73 43 | -38 36 | 1939 | 65, 440 | 25, 125 | 123 |
| S2-1985 | 1985 | -74 40 | -33 56 | 545 | 371, 283, 258 | 25, 100, 190 | -10* |
| D1 | 1985 | -74 04 | -35 45 | 2100 | 352 | 25, 100 | 13* |
| D2 | 1985 | -74 15 | -35 22 | 1800 | 281, 52 | 25, 100 | 70* |
| S2-1987 | 1987 | -74 40 | -34 00 | 558 | 352, 407 | 25, 100 | -10* |
| S3 | 1992 | -74 35 | -32 39 | 659 | 165, 356 | 70, 170 | -10* |
| Fr1 | 1995 | -75 01 | -31 46 | 610 | 691, 837, 828, 828 | 20, 126, 232, 353 | 50* |
| Fr2 | 1995 | -75 02 | -33 33 | 574 | 683, 837, 829, 829 | 20, 126, 232, 383 | 25 |
| F1 | 1998 | -74 31 | -36 36 | 647 | 327, 277, 393 | 10, 56, 207 | 124 |
| F2 | 1998 | -74 25 | -36 22 | 1180 | 309, 390, 326, 348 | 10, 56, 202, 433 | 106 |
| F3 | 1998 | -74 17 | -36 04 | 1637 | 395 | 56, 413 | 94 |
| F4 | 1998 | -74 09 | -35 42 | 1984 | 376, 390, 332 | 10, 56, 207 | 90 |
| S2-2003 | 2003 | -74 40 | -33 28 | 596 | 421 | 25, 100 | -10* |
| M1 | 2009 | -74 13 | -32 19 | 967 | 365 | 25, 46 | 110 |
| M2 | 2009 | -73 58 | -32 16 | 1898 | 364 | 19, 78:4:150 | 110 |
| M3 | 2009 | -74 30 | -30 09 | 725 | 361 | 25, 123:4:199, 310:5:505 | 110 |
| M4 | 2009 | -74 26 | -30 02 | 1051 | 361 | 25, 442:16:986 | 110 |
| M5 | 2009 | -74 10 | -29 32 | 1917 | 361, 336 | 26, 55:16:391 | 110 |
| S2-2010 | 2010 | -74 38 | -33 30 | 612 | 363 | 25, 104, 176 | -10* |
| W2 | 2010 | -74 23 | -36 01 | 1411 | 361, 344, 302, 318 | 25, 84, 194:4:234, 289:4:389 | 94 |
| W3 | 2010 | -74 13 | -35 55 | 1488 | 363, 363, 91, 304 | 25, 93, 163:2:209, 216:4:272 | 97 |
| SB | 2013 | -77 00 | -34 28 | 705 | 371 | 51:8:395 | 37 |
| SC | 2013 | -77 45 | -36 09 | 700 | 376 | 26:4:214 | 28 |
| SD | 2013 | -77 00 | -34 03 | 505 | 371 | 19:4:119 | 37 |
| SE | 2013 | -77 01 | -34 14 | 590 | 196 | 175 | 37 |

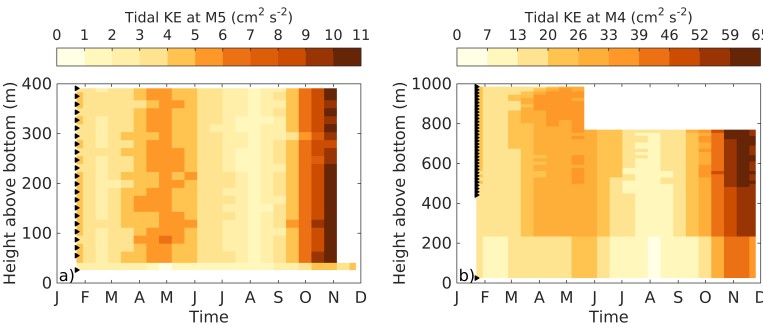

**Figure 4.** Diurnal tidal KE over time and depth at a) mooring M5 and b) mooring M4 on the continental slope, located above the 1900 m and 1050 m isobath, respectively. Black triangles mark the depths of individual measurements. Note the different scale for diurnal tidal KE and vertical axis in a) and b).

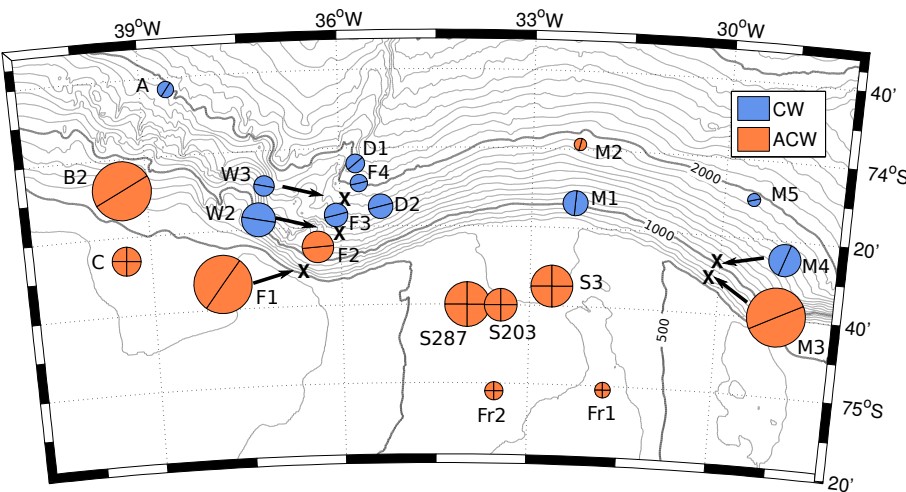

**Figure 5.** Map showing the maximum depth-averaged, diurnal tidal KE reached in austral summer. The area of the circles is proportional to the square root of the KE. Blue circles indicate a clockwise (CW) rotation of tidal currents at the $K_1$ frequency, red circles an anticlockwise (ACW) rotation. The line through the circles indicates the orientation of the major axis (i.e. strongest tidal current). Tidal current ellipses with an aspect ratio (semi-minor/semi-major axis) of more than 0.8, i.e. close to circular, are marked with two crossing lines instead of a single line. At the location of mooring S2 with five years of measurements, the data for the deployment years 1987 (S287) and 2003 (S203) are presented. Some circles are displaced from their actual mooring locations (marked with "X") for legibility.

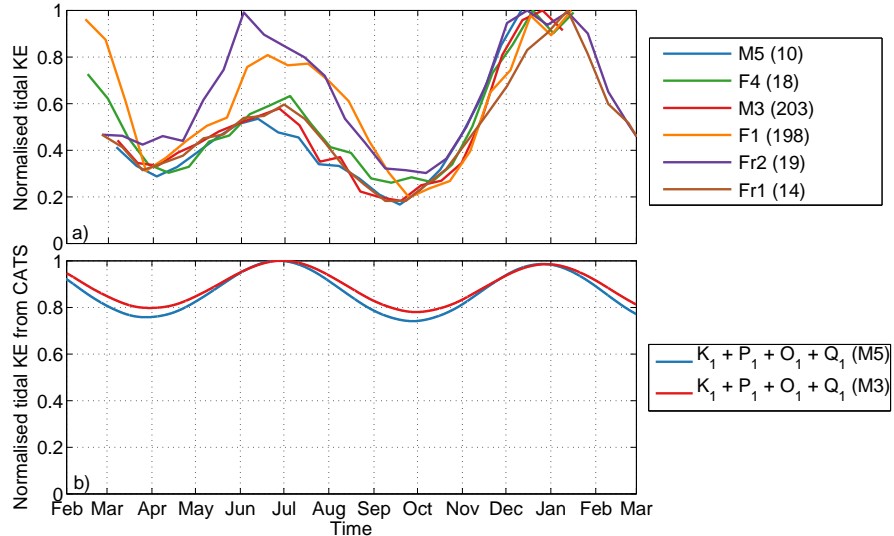

**Figure 6.** a) Time series of normalised diurnal tidal KE at moorings M5 and F4 (deeper continental slope), M3 and F1 (shelf break) and the first year of record at Fr2 and Fr1 (shelf). At M5, KE is only calculated until the ADCP stops measuring (see Table 1). The maximum diurnal tidal KE values in $\mathrm{cm^2\,s^{-2}}$ are given in parentheses in the legend. b) Time series of normalised tidal KE at the $K_1+P_1+O_1+Q_1$ tidal frequencies as predicted by the CATS tidal model at the locations and deployment times of moorings M3 and M5.

**Table 2.** Parameters of the reference stratification profile and corresponding ranges of change in the sensitivity test (SM: surface magnitude, SSM: subsurface magnitude, SSD: subsurface depth, DM: deep magnitude). For case DM, the average (av.) is given in addition to the range of values in the profile section.

| | SM, $10^{-5}\,\mathrm{s^{-2}}$ | SSM, $10^{-5}\,\mathrm{s^{-2}}$ | SSD, m; model level | DM, $10^{-7}\,\mathrm{s^{-2}}$ |
|---|---|---|---|---|
| reference stratification | 3.20 | 0.17 | 640; 5 | 1.99–4.42; av.: 2.92 |
| sensitivity test | 0.07–4.05 | 0.04–2.78 | 320–960; 3–7 | 1–30 |

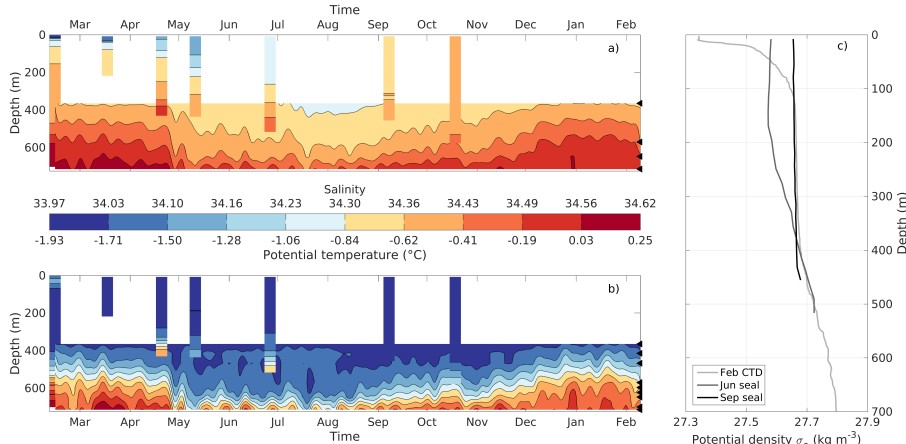

**Figure 7.** Hovmöller diagrams for a) salinity and b) temperature. The continuous records below approximately 350 m depth are the hydrographic time series of moored instruments at M3, acquired in 2009 (see Table 1). The mooring data have been low-pass filtered by applying a fourth order Butterworth filter removing variability at shorter periods than the cut-off period of 168 h (one week). Black triangles mark the depths of individual measurements. CTD profiles from ship (from the deployment cruise of mooring M3 in 2009, first profile) and seals (obtained in 2011 from within approximately 10 km distance to mooring M3, see Fig. 1) complement the mooring records. The width of the profiles is arbitrarily set to one week for clarity. Panel c) shows the seasonal development of potential density based on three of the temperature and salinity profiles near mooring M3.

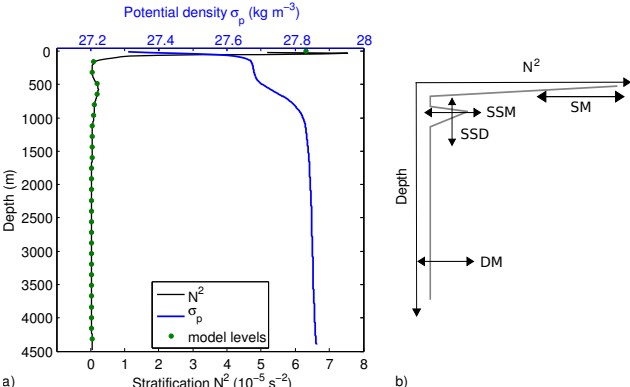

**Figure 8.** a) Profiles of potential density (blue line) and resulting stratification ("reference stratification", black line) from historic CTD data obtained in January and February in the area of moorings M1 to M5, merged with profiles from the deeper Weddell Sea at depth. Green markers indicate the vertical levels of the numerical code. b) Sketch showing the parameters changed in the stratification sensitivity tests. Arrows indicate direction (but not magnitude) of changes. SM: surface magnitude, SSM: subsurface magnitude, SSD: subsurface depth, DM: deep magnitude.

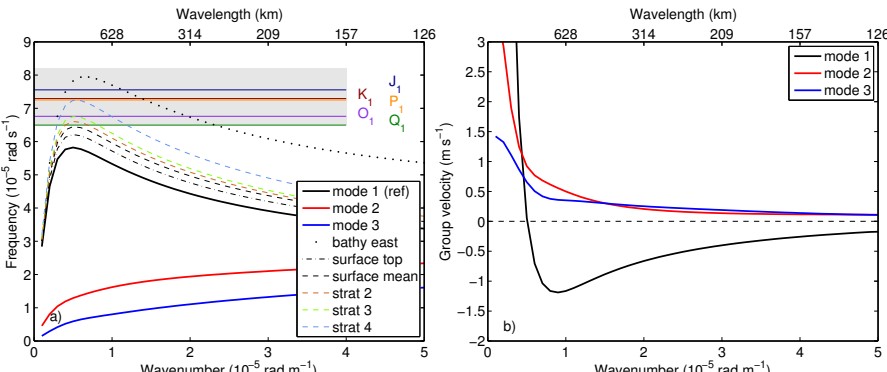

**Figure 9.** a) Dispersion curves for CTWs of modes 1–3 using the reference stratification (thick lines). In addition, the dispersion curves for mode 1 CTWs for the bathymetry east of the study area ("bathy east"), differently inferred surface $N^2$ values for the reference stratification ("surface top", using the uppermost value of the observational stratification profile, and "surface mean", using the average of the upper 80 m of the observed stratification profile) and three other stratification profiles are shown. Stratification profiles 2–4 are representative for regions in the study area with increasing distance west of the reference stratification and inferred similarly. The diurnal tidal band is shaded in light grey with the most important diurnal frequencies marked by coloured lines. b) Group velocities for CTWs of modes 1–3 using the reference stratification. Zero group velocity is indicated by a dashed line.

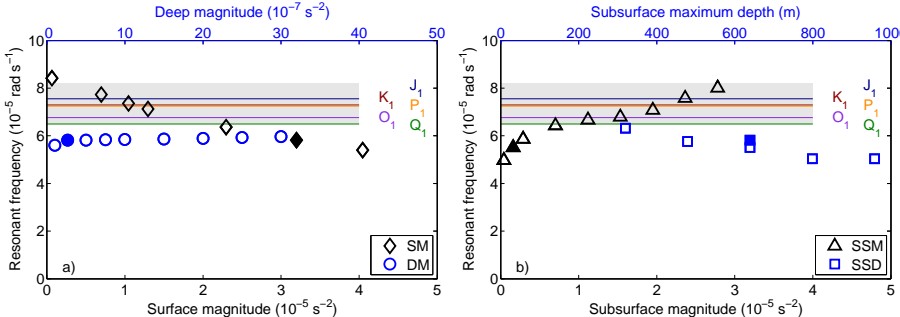

**Figure 10.** Sensitivity test for changing stratification. a) The surface magnitude (SM) and deep magnitude (DM) are varied. b) The subsurface magnitude (SSM) and its depth (SSD) are varied. Results from the reference stratification profile are indicated by filled markers. For case SSD, the shape of SSM in the profile is simplified (open marker below filled marker) and then varied in depth. The diurnal tidal band is shaded in light grey with the most important diurnal frequencies marked by coloured lines.

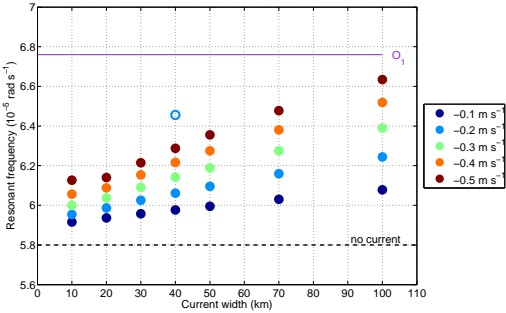

**Figure 11.** Sensitivity test for changing the width and strength of the slope current. Additionally, an example for the RF of a less barotropic (i.e. surface-enhanced) current is shown by the open blue marker. The RF for the reference stratification without added current (see Fig. 9) is indicated by a broken black line, and the tidal frequency $O_1$ is marked by a solid purple line.

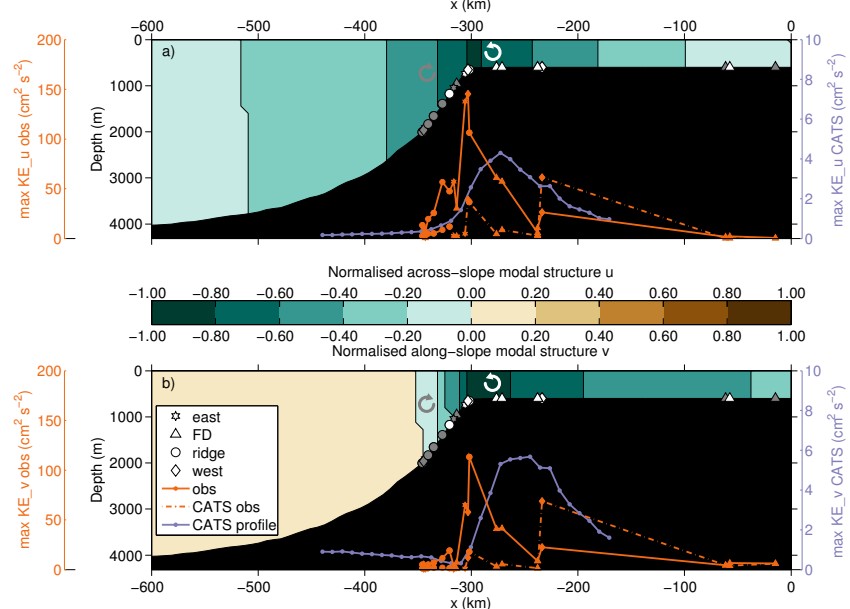

**Figure 12.** Across-slope sections of the normalised a) along-slope ($u$) and b) across-slope ($v$) modal structure from the numerical code (background colours) at the wave number of the RF for reference stratification. Moorings within the modelled domain are indicated by markers whose shapes correspond to the geographic locations (west, ridge, Filchner Depression, east) as in Fig. 1. Markers filled in grey (white) indicate a clockwise (anticlockwise) rotational sense of the diurnal tidal currents. For each mooring location, diurnal tidal KE for austral summer is shown, both from observations (solid orange line) and derived from CATS (broken orange line). Diurnal tidal KE predicted from the CATS run along the cross-shelf section through the locations of moorings M1 and M2 (see Fig. 1 for location) is shown as purple line with markers indicating locations of predicted currents.

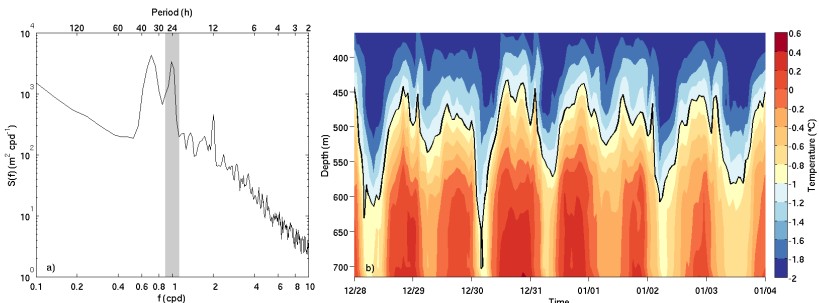

**Figure 13.** a) Power spectral density of the height above the bottom of the -1° C isotherm which indicates the transition between WW and WDW at M3. The diurnal tidal frequency band is marked in grey. b) Hovmöller diagram of low-passed filtered temperature at mooring M3 from 28 December to 4 January. The -1° C isotherm is shown by a black line.

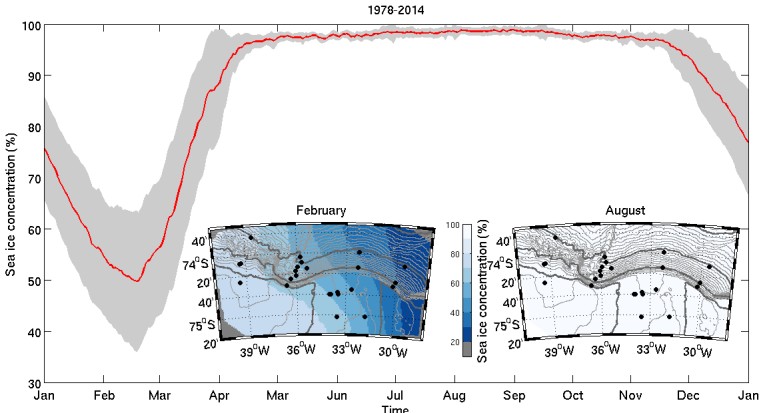

**Figure 14.** Sea ice concentration averaged over the study area and the period 1978-2014 (red line) with standard deviation (grey). Insets show maps of mean sea ice concentrations in February and August; isobaths and mooring locations (black dots) are as in Figure 1.