# Peer review of "Seasonal resonance of diurnal coastal trapped waves in the southern Weddell Sea, Antarctica"

_Ocean Science, 2016_

## Referee Comment (RC1) · Anonymous Referee #1 · 23 Jun 2016

Review of "Seasonal resonance of diurnal coastal trapped waves in the southern Weddell Sea, Antarctica" By S. Semper and E. Darelius

The authors provide observational evidence for large amplitude diurnal tides on the Weddell Sea slope, and then they seek to rationalize this finding in terms of resonant (zero group velocity) coastal-trapped waves. They go on to demonstrate the conditions (notably mean flow and stratification) that can make the resonant frequency vary from time to time. Overall, this is a credible piece of work, although there are perhaps some places where a bit more could be done.

The main extension I see would be to look at records from the same (or nearly so) isobath, but separated alongshore. These pairs can then be used to estimate alongshore wavelengths. This makes the most sense when moorings are simultaneous, but even if they are not, differences in Greenwich phase could be used to see if the estimated wavelength is at least the right magnitude. Also, the model results provide other information, such as direction of current vector rotation and amplitude of current components, that could also be used to compare with observations. Maybe all we will learn is that things have the right magnitude, but it could be more rewarding.

I recall reading somewhere that a PhD thesis by M. Spillane (Oregon State, 1980) showed that the wave dispersion curves can do strange things when inviscid group velocity vanishes. However, I do not know of anything quite like this is the normal literature. It would be interesting to know more about this but it would be asking too much to have this effect covered in this submission.

Some specific points - Line 37: Given the mooring notation, it is particularly important here to use the correct (subscripted) tidal notation consistently, e.g., is this $M_2$ or M2? (Yes, it IS clearly done right in other places). - 95: Please say more about the CATS tidal model. Is it barotropic? Does it include non-tidal currents? Is it nonlinear? Etc. - 121: Say here how thick the boundary layer is. - Page 5: Why normalize tidal KE? You are throwing out useful information (actual amplitude), and I do not see any advantage for normalizing. - 170: The evidence here on ambient currents vis a vis waves is pretty weak. True, you can say that the "mean" alongshore currents vary with time, but, evidently useful local information about the time dependence and the spatial structure is lacking. There is nothing you can do about this, of course, but it would be well to advise the reader that the main thing you can glean here is the magnitude of the "mean" flow, and that it (probably) does vary from time to time. - 243-244: This sentence does not make sense to me. - 264: I am not sure what is meant by a dispersion curve showing the waves to be barotropic. There must be a missing step in the argument here. Maybe they mean the modal structure (Fig 11) is barotropic? Also, give a representative range of Bu. It is probably a wording issue, but the last sentence of this paragraph seems to contradict the preceding text. - 270: Enhanced relative to

what? - 272: What asymmetry is that? - Figure 7: Label axes.

Again, I believe that this contribution is sound overall, but that improvements ought to be made.

---

## Referee Comment (RC2) · L. Padman (Referee) · 24 Jun 2016

This paper is a comprehensive assessment of the seasonal variability of diurnal tidal currents in the southern Weddell Sea, using data from 29 moorings between 1968 and 2014. While the seasonal variability is known, and has been previously interpreted as response of diurnal tide-forced shelf waves to changing stratification and along-slope current (various papers from the 1980's), the increased data base and interpretation through sensitivity tests on an idealized model (Brink, 2006) makes this a valuable new study.

My specific comments are provided as margin comments and edits on the marked *.tex file. Most of these are relatively minor. Some major comments are as follows:

[Figure]

1. Lines 43-55, about CTWs, will possibly make no sense to a lot of readers without a dispersion curve to look at. Probably this means adding a simple sketch of one, showing that cp is always with shallow water on the left (in the Southern Hemisphere), then three cases of cg (+ve, -ve, and 0 (RF)).

2. Much of the Discussion is actually Introduction (Background) material. Move everything you knew, or should have known, before the study into Introduction. Discussion could keep some implications (regarding sea ice, mixing etc) that depend on the magnitude of the results you have presented, but expectations should be set in Introduction.

3. The discussion of Figure 5 is not very clear. I think the argument you are trying to make is that the "diurnal band" as a whole shows mainly semi-annual, which you ascribe to K1/P1 modulation. Then, K1 (removing P1 influence by inference) is "annual". But the modulation of the diurnal band at the semi-annual frequency is too large for K1+P1, even without the other tidal lines (e.g., O1) being caught in the definition of "diurnal band". Some more thought about this, and an improved discussion, would be useful. It is not impossible that currents and stratification add to a semi-annual term in CTW properties.

4) What does it mean to have a wave whose wavelength ($\sim$1300 km) is an order of magnitude longer than typical along-slope scales of isobath variability?

– Laurie Padman

Please also note the supplement to this comment:
http://www.ocean-sci-discuss.net/os-2016-36/os-2016-36-RC2-supplement.pdf

[Figure]

**Supplement:**

**General Statement**

Review of "*Seasonal resonance of diurnal coastal trapped waves in the southern Weddell Sea, Antarctica*" By S. Semper and E. Darelius

This paper is a comprehensive assessment of the seasonal variability of diurnal tidal currents in the southern Weddell Sea, using data from 29 moorings between 1968 and 2014. While the seasonal variability is known, and has been previously interpreted as response of diurnal tide-forced shelf waves to changing stratification and along-slope current (various papers from the 1980's), the increased data base and interpretation through sensitivity tests on an idealized model (Brink, 2006) makes this a valuable new study.

My specific comments are provided as margin comments and edits on the marked *.tex file. Most of these are relatively minor. Some major comments are as follows:

1. Lines 43-55, about CTWs, will possibly make no sense to a lot of readers without a dispersion curve to look at. Probably this means adding a simple sketch of one, showing that cp is always with shallow water on the left (in the Southern Hemisphere), then three cases of cg (+ve, -ve, and 0 (RF)).

2. Much of the Discussion is actually Introduction (Background) material. Move everything you knew, or should have known, before the study into Introduction. Discussion could keep some implications (regarding sea ice, mixing etc) that depend on the magnitude of the results you have presented, but expectations should be set in Introduction.

3. The discussion of Figure 5 is not very clear. I think the argument you are trying to make is that the "diurnal band" as a whole shows mainly semi-annual, which you ascribe to K1/P1 modulation. Then, K1 (removing P1 influence by inference) is "annual". But the modulation of the diurnal band at the semi-annual frequency is too large for K1+P1, even without the other tidal lines (e.g., O1) being caught in the definition of "diurnal band". Some more thought about this, and an improved discussion, would be useful. It is not impossible that currents and stratification add to a semi-annual term in CTW properties.

4) What does it mean to have a wave whose wavelength *1300 km) is an order on magnitude longer than typical along-slope scales of isobaths variability?

-- Laurie Padman

Abstract

The summer enhancement of diurnal tidal currents at the shelf break in

the southern Weddell Sea is studied using velocity measurements from 29

moorings during the period 1968 to 2014. Kinetic energy associated with

diurnal tidal frequencies is largest at the shelf break and decreases rapidly with distance, and its magnitude increases from austral winter to summer by, on average, 50\,\%. The summer enhancement is observed in all deployments.

The observations are compared to results from an idealised numerical solution of the properties of coastal trapped waves (CTWs) for a given bathymetry, stratification and an along-slope current. The frequency at which the dispersion curve for mode 1 CTWs displays a maximum (i.e.\ where the group velocity is zero and resonance is possible) is found within or near the diurnal frequency band, and it is sensitive to the stratification in the upper part of the water column and to the background current. The maximum of the dispersion curve is shifted towards higher frequencies, above the diurnal band, for low stratification and a strong background current (i.e.\ winter-like conditions) and towards lower frequencies for strong upper layer stratification and a weak background current (summer).

The seasonal evolution of hydrography and currents in the region is inferred from available mooring data and conductivity-temperature-depth profiles.

Near-resonance between CTWs and the diurnal tides during austral summer can explain the observed seasonality in tidal currents.

\introduction | %% \introduction[modified heading if necessary]

**Commented [LP1]:** The tide-forced CTW's *are* tides; so maybe better here to say "Near-resonance of diurnal tidal CTWs during …"

**Commented [LP2]:** I think the Introduction needs to include Nicholls et al., 2009 Reviews of Geophysics.

Also, a lot of Discussion should really be moved to Introduction. Anything relevant that you knew or should have known before starting the study should be in the Introduction.

The shelf break region in the southern Weddell Sea (Fig.~\ref{fig:map})
is an area of great climatic interest.
This is where cold, dense Ice Shelf Water emerging from underneath the
Filchner-Ronne Ice Shelf (FRIS) descends the continental slope
\citep{Foldvik2004}, ultimately contributing to the formation of
Antarctic Bottom Water which spreads out into the major oceans at abyssal
depths \citep{Orsi1999}.

Furthermore, warm off-shelf water crosses the shelf break during summer

\citep{Arthun2012} and flows southward towards the Filchner Ice Shelf
along the eastern flank of the Filchner Depression \citep[][see map in
Fig.~\ref{fig:map} for location]{Foldvik1985d}. The wind driven inflow of
Modified Warm Deep Water (MWDW) has been observed to reach the ice shelf
front
\citep{Darelius2016}, and climate models suggest a larger inflow and a
dramatic increase in basal melt rates below the FRIS within the next
century \citep{Hellmer2012}.

Physical processes at the shelf break and on the continental slope
influence both the cold outflow and the warm inflow, and to some extent
set their hydrographic properties and strengths. The variable depth of

the thermocline, for example, which is controlled mainly by wind forcing
and eddy overturning \citep{Sverdrup1953, Nost2011} will determine if and
when warm water can access the continental shelf \citep{Arthun2012}.
Meanwhile, it affects the density contrast between the cold outflow and
the ambient water at the shelf break, and thus the strength of the
geostrophically balanced outflow \citep{Kida2011, Wang2012} as well as
the properties of the descending dense plume since it is a mixture of

outflow water and ambient water, \citep{Darelius14_Makinson}. The co-location of the critical latitude for the tidal component M2 and a critical slope leads to enhanced turbulence levels in the region \citep{Fer2016}. Mixing can be expected to be further enhanced at the shelf break by the strong diurnal tidal currents and the presence of continental shelf waves \citep{Middleton1987, Foldvik1990, Jensen2013}, a class of coastal trapped waves (CTWs).

This study focusses on tidally generated CTWs at diurnal frequencies in the shelf-break region of the southern Weddell Sea.

CTWs can be generated by e.g.\ tides \citep{Thomson1982} or wind \citep{Huthnance1995}.

Additionally, a connection between the generation of the waves and the outflow of dense shelf water through troughs has been suggested \citep{Marques2014, Jensen2013}.

CTWs with sub-inertial frequencies propagate along a trapping boundary, e.g.\ a coastal wall or a sloping bottom \citep{Huthnance1995, Huthnance1986}. The waves require the support of such a boundary to exist and decay exponentially with increasing distance from it \citep{Mysak1980}.

While CTWs propagate with shallow water to the left (in the southern hemisphere), the group velocity $c_g$, and thus the energy associated with the waves, can propagate in either direction.

If the group velocity is zero, i.e.\ for a maximum in the dispersion curve of a wave, energy cannot propagate. When the frequency of this maximum (hereafter called ``resonant frequency'', RF) in the dispersion

**Commented [LP5]:** Note sure I got the TeX right, but just include Darelius as a regular cite.

**Commented [LP6]:** You need to explain the source of the mixing better here. I think you are referring to bottom stress. If you want to claim a baroclinic source of mixing, then needs a cite. Fer et al. paper in prep. would be good, but maybe Ilker's Yermak Plateau paper?

**Commented [LP7]:** I think this discussion will not make much sense to most readers until you show a dispersion curve for a CTW, even if it is just a schematic showing cp (always shallow water on left), and the three cases of cg>0, cg<0, and RF.

relation coincides with the frequency of tidal currents, resonance may occur and tidal currents will be amplified.

%%%%%%%%%%%%%

%after editor comment

In practice, it is likely that some energy can escape on one side along the shelf, resulting in near-resonance rather than resonance.

%%%%%%%%%%%%%

Commented [LP8]: This expression isn't clear; probably need a little introduction to bathymetric irregularity, convergence etc.

Such near-resonant diurnal CTWs were first recorded on the shelf of the Outer Hebrides of Scotland by \cite{Cartwright1969} and have been observed and modelled at numerous occasions and locations since then \citep[e.g.\ ][]{Huthnance1974, Crawford1982, Heath1983, Hunkins1986, Padman1992, Skardhamar2015}.

In our study region, \cite{Foldvik1974} and \cite{Foldvik85_bottom_currents} first suggested that CTWs caused the observed strong diurnal tidal currents, which broke down during winter presumably due to a seasonally varying stratification.

Later, \cite{Middleton1987} and \cite{Foldvik1990} found a particularly strong enhancement of the K$_1$ tidal constituent during austral summer. The summer maximum was hypothesised to be due to the interaction of barotropic CTWs with topography in the presence of a seasonally variable mean current \citep{Foldvik1990}.

These studies were based on a small number of moorings and a barotropic shelf wave model neglecting the effects of stratification.

Our study is based on a more extensive data set and aims to provide new insight into the seasonal variability of the tidal currents at diurnal frequencies and its causes in the southern Weddell Sea.
Observations of current velocities from 29 moorings are used to quantify the strength of diurnal tidal currents and to describe their spatial and temporal variability.
We provide a novel description of the seasonal changes in shelf break hydrography based on observations and use a numerical code
\citep{Brink2006} in order to investigate the sensitivity of the CTW properties to seasonal changes in hydrography.
The effects of the stratification and slope current on CTWs are compared, and the influences of the bathymetry and sea ice on CTWs are discussed.

\section{Data and methods}
\label{sec:methods}

%mooring data
Current meter data from 29 moorings
\citep{Foldvik2004,Jensen2013,Darelius2016} located on the continental slope and shelf in the area surrounding the Filchner Depression have been analysed. The records span the years 1968 to 2013 with each being of 1--2 years duration. The locations of the moorings are shown in
Fig.~\ref{fig:map}, and deployment details are listed in
Table~\ref{tab:moorings}.

%coord. system rotation

The coordinate system is rotated clockwise to align the $y$-axis with the isobaths, agreeing with the set-up of the numerical code \citep{Brink2006} in the southern hemisphere. $u$ is thus directed on-shelf and $v$ along the continental slope (Fig.~\ref{fig:map}). The rotation angle $\beta$, positive for clockwise rotation, is listed in Table~\ref{tab:moorings}; it is inferred for each mooring from the local bathymetry based on the GEBCO\_2014 bathymetry grid (The GEBCO\_2014 Grid, version 20150318, \url{http://www.gebco.net}) with an estimated accuracy of approximately $\pm$10$^\circ$.

%spectral analysis
Time series of kinetic energy (KE) associated with the diurnal tidal currents are constructed as follows: The hourly-averaged current meter data are divided into overlapping chunks of 1.5\,months length beginning every 14th day. For each chunk, the power spectral densities are estimated using Welch's method \citep{Welch1976} and three 50\,\%-overlapping Hanning windows.
%KE
The diurnal tidal KE is obtained by integrating the velocity spectra,
\begin{equation}
\text{KE} = \int^{\omega_{2}}_{\omega_{1}}(S_{u}+S_{v})\,d\omega,
\end{equation}
%where %$u'=u-\bar{u}$ and $v'=v-\bar{v}$ denote the velocity fluctuations, the overbar indicates time averaging, and
%$S_{u}$ and $S_{v}$ are the power spectral densities of $u$ and $v$,
respectively.

**Commented [LP9]:** This approach is sensible, but relies on choosing a length scale for the calculation of local isobaths orientation.

**Commented [LP10]:** Better word than "chunk" ? Maybe "intervals" ?

**Commented [LP11]:** So, how long is each window? I think this needs ¾ of a month to work, but that is a bad window length for tides. Better to use 14 or 29 days (more precisely, the spring/neap for O1/K1)

where, following \cite{Jensen2013}, $\omega_{1}$ and $\omega_{2}$
correspond to periods of 26.9\,h and 21.3\,h respectively.
%The results presented are not sensitive to small changes in w1 and w2
(MAKE SURE THAT THEY ARE NOT!)

%CATS
Diurnal tidal KE has also been inferred using tidal predictions from the
Circum-Antarctic Tidal Simulation \citep[CATS, ][]{Padman2002} for the
respective time and location of every mooring deployment. The tidal
predictions are treated in the same way as the observational current
velocities.

%T_TIDE
Tidal ellipses have been obtained from the mooring records using harmonic
analysis \citep[T\_TIDE, ][]{Pawlowicz2002}, a Matlab version of the
FORTRAN code developed by Foreman (1978).
When evaluating seasonal changes, the harmonic analysis was carried out
on 50\,\% overlapping month-long segments of the records. The tidal
constituent P$_1$ was then inferred from K$_1$ based on the year-long
record, since the month-long segments are too short to separate the two
signals.

%OTHER DATA
%deployment CTD and seals
Records of temperature and salinity from mooring M3, located at the
725\,m isobath just to the east of the Filchner Trough sill
(Fig.~\ref{fig:map}), are used to  describe the seasonal changes in

hydrography at the shelf break and upper continental slope. The mooring
records are complemented by a conductivity-temperature-depth (CTD)
profile obtained during the deployment cruise in 2009 and by hydrographic
measurements obtained in the vicinity of the M3 location (within 10\,km,
Fig.~\ref{fig:map}) provided by seals tagged with small CTD sensors
\citep[described in][hereafter referred to as "seal data"]{Arthun2012}.
The accuracies of the seals' temperature and salinity measurements are
stated to be 0.005$^\circ$\,C and 0.02, respectively \citep{Boehme2009}.

%sea ice and Halley
In addition, we use wind observations from Halley Research Station,
located at 75$^\circ$\,35'\,S, 26$^\circ$\,39'\,W (Fig.~\ref{fig:map}),
from 1957 to 2014 \citep{BAS2013} and satellite derived records of sea
ice concentration \citep{Meier2013}, available for the period 1978 to
2014. The sea ice concentration is averaged over the study area (inset in
Fig.~\ref{fig:map}).

\section{Observational results}

\subsection{Spatial and temporal variability of tidal currents}

%spectral analysis
The diurnal tidal frequency band shows enhanced variance for both the
$u$- and $v$-component, especially at the frequencies of the most
important diurnal tidal constituents K$_1$ and O$_1$
(Fig.~\ref{fig:psd}).

High energy levels are additionally observed at semi-diurnal frequencies
and around 35\,h, as reported by \cite{Jensen2013}.

%KE
The energy associated with the diurnal tidal currents, the diurnal tidal
KE (Sect.~\ref{sec:methods}), shows little variation with depth, except
for a boundary layer at the bottom where diurnal tidal KE is slightly
decreased compared to the overlying water column
(Fig.~\ref{fig:hovmollerEKE}). Depth-averaged diurnal tidal KE is used
for further analysis.

%KE pie map
Figure~\ref{fig:EKEpiemap} shows the spatial distribution of diurnal
tidal KE during austral summer. The magnitude of diurnal tidal KE is
highest directly at the shelf break (e.g.\ moorings B2, F1, M3) and
decreases rapidly with distance from it.
The tidal currents rotate clockwise on the deeper continental slope and
anticlockwise at the shelf break and on the shelf.
The major axes of the tidal ellipses at the K$_1$ frequency are directed
across the continental slope for moorings located at the shelf break and
on the continental slope in the eastern part of the study area, while
tidal currents recorded at moorings on the shelf are close to circular.

%seasonality, explain K1-P1-interference
Time series of diurnal tidal KE (Fig.~\ref{fig:ttidecurrents}a) show two
local maxima; one in austral summer and one in austral winter.

These two peaks per year result from the interference of the diurnal
tidal constituents K$_1$ and P$_1$, which are in phase every six months.
The austral summer maximum is 30\,\% to 180\,\% higher than the winter
maximum. This asymmetry is especially strong in records from moorings on
the continental slope and at the shelf break, but it is observed in all
deployments of sufficient length.

For moorings on the continental shelf, the difference between the maxima
is sometimes less pronounced (e.g.\ Fr2 in
Fig.~\ref{fig:ttidecurrents}a).
Minima of the diurnal tidal KE occur near the equinoxes in spring and
autumn, when both the sun and the moon are close to the equator.

%normalised t_tide currents
Time series of the K$_1$ magnitude obtained from harmonic analysis on
monthly segments (see Section \ref{sec:methods}) show a seasonal signal
with a maximum during austral summer which is apparent at all moorings
(Fig.~\ref{fig:ttidecurrents}b) with the exception of F1, located at the
647\,m isobath downstream of the Filchner outflow. F1 shows no increase
in magnitude towards the end of the record when approaching austral
summer.
For all moorings, the semi-major axes are largest in the across-slope
component $u$.

%%%%%%%%%%%%%%%%%%%%%%%%%%%%%%%%%%%%%%%%%%%%%%%%%

\subsection{Seasonal variability of the hydrography and current on the
upper slope}

**Commented [LP20]:** Something is wrong here! P1 and K1 don't explain this high degree of semiannual variability, especially as the "diurnal band" still contains O1.

\label{sec:hydrography}

%T&S - moor&seal combined plot
The seasonal variability in the hydrography at the shelf break and on the upper slope is investigated by merging all available observational data (moorings, CTD, seal data) near the location of mooring M3 (Fig.~\ref{fig:sealmoor}a,b).

%
Cold and fresh Winter Water (WW) is found on top of warm and saline MWDW. MWDW is a mixture of WW and Warm Deep Water (WDW), the Weddell Sea version of the Circumpolar Deep Water which is the major component of the Antarctic Circumpolar Current.

While the temperature in the upper approximately 400\,m is near the freezing point year-round, the salinity of the surface layer increases from 34.0 in February to 34.4 in October. The cold and fresh surface layer during summer likely results from local sea ice melt.

The thermocline is found at a depth of approximately 400\,m from December to April and deepens by 200\,m to approximately 600\,m during May to August. Seasonal changes in the water column below the pycnocline are negligible.

%
Generally, the seal data show higher salinities and temperatures at depth compared to the mooring data (also compared to the range of the unfiltered mooring records, not shown), suggesting that

Commented [LP21]: This is a nice figure. However, is it really a "Hovmoller" diagram as stated in the caption?

https://en.wikipedia.org/wiki/Hovm%C3%B6ller_diagram

Commented [LP22]: I disagree: In your Figure 6, if I plotted T and S at the bottom of the plot, I would see a seasonal signal.

the MWDW and the thermocline are found higher up in the water column in 2011 compared to 2009. %Differences in seasonal inflow of MWDW onto the shelf have been discussed by \cite{Arthun2012} and \cite{Darelius2016}.

[revised manuscript text omitted]

Commented [LP24]: If the wavelength is really this long, does it make sense to think of these as 'waves' when along-slope topography varies on much smaller scales?

i.e., Discuss implications for wavelength >> topo scales

Commented [LP25]: It's been so long since you mentioned this, you might need to explain it again.

Commented [LP26]: So … why not use higher resolution in the Brink model? This is the obvious thing to do. I know there are some stability issues with this code, and maybe that's the answer. But if this is the reason, you need to discuss it here, or maybe better in Section 4.1.

value or an average of the upper 80\,m shifts the dispersion curves and

thus the RF to higher frequencies (Fig.~\ref{fig:dispcurve}).

%

%other ref profiles

For stratification profiles which are representative for areas farther

west at the shelf break and constructed similarly to the reference

stratification with surface $N^2$ values of the upper 80\,m average, the

dispersion curve and RF are similarly shifted to higher frequencies

(Fig.~\ref{fig:dispcurve}).

Keeping in mind the variations along the shelf break and with different

approaches on how to choose the uppermost stratification value, the

characteristic parameters of the reference profile (SM, SSM, SSD, DM,

Fig.~\ref{fig:refstrat}b) are varied in the following to explore the

general effects of stratification on the dispersion curve and the RF.

%stratification test

Figure~\ref{fig:strattest} shows the results from the sensitivity test

for stratification, where the RF is identified from each dispersion curve

obtained from the modified stratification input. An increase of $N^2$ at

the surface (case SM) leads to a decrease in RF, which moves through the

diurnal tidal frequency band for the modelled range of surface

stratification. In contrast to case SM, an increase of the stratification

maximum at approximately 640\,m depth (case SSM) increases the RF.

The effect of an increase in depth of the subsurface maximum (case SSD)

results in an apparent decrease of the RF. However, due to the

interpolation in the numerical code, the stratification around the

subsurface maximum as well as the exact value of the maximum are difficult to preserve. Hence, the actual effect of case SSD appears to be rather small.
Varying the stratification below 1200\,m depth (case DM) has a negligible effect on the RF.

%%%%%%%%%%%%%%%%%%%%%%%%%%%%%%%%%%%%%%%%%%%%%%%%%%%

%current sensitivity test
\subsection{Sensitivity to along-slope current} %Merge maybe with stratification section

%current parameters
The optional along-shore current has a Gaussian shape; its offshore, onshore, upward and downward $e$-folding length scales must be specified, in addition to the centre position, strength and depth of the current.

For the sensitivity test, a barotropic (i.e.\ with a large vertical length scale) westward current is assumed which is centred at the shelf break.
The density is set to be undisturbed at the coast when the density field is altered according to the thermal wind equation, with the input $N^2$ vector being the reference stratification for all runs.
The width and strength of the current are varied from 10 to 100\,km and 0.1 to 0.5\,m\,s$^{-1}$, respectively (Fig.~\ref{fig:currenttest}).
Generally, both a stronger and a wider current lead to an increase in RF; with the effect of strength being largest.

Commented [LP27]: Figure 10 needs O1 and K1 frequency lines marked.

Moving the location of the current core 40\,km on (off) shore, the
sensitivity of the RF is increased (reduced) slightly compared to a
current core at the shelf break. The magnitude of change in RF equals
approximately a change in current velocity of $\pm10\,cm\,s$^{-1}$ (not
shown).

Although the overall effect of an added barotropic slope current is minor
compared to the sensitivity to changes in stratification (cp.\ $y$-axes
in Fig.~\ref{fig:strattest} and Fig.~\ref{fig:currenttest}), the
sensitivity depends noticeably on the vertical length scale. As an
example, a 40\,km wide and 0.2\,m\,s$^{-1}$ fast current is chosen and
its downward $e$-folding length scale is reduced from 4300\,m to 2000\,m.
The RF is then considerably larger (open circle in
Fig.~\ref{fig:currenttest}) than for the more barotropic case.

\section{Discussion}

Observations from the continental slope in the southern Weddell Sea show
anomalously strong tidal currents at diurnal frequencies
\citep{Middleton1987}. Our extended analysis - including all current
meter records (1968--2014) from the region - confirms previous findings
suggesting that the strong currents are the result of tidally forced CTWs
\citep{Middleton1987,Foldvik1990,Foldvik85_bottom_currents}. The
observations agree qualitatively with the mode 1 CTW "generated" in the
numerical code by \cite{Brink2006}, Fig.~\ref{fig:allplot}; notably the

Commented [LP28]: I do not like this bracketed way of doing opposite cases. In general, you can ignore the bracketed examples as they are implied as the opposite of the main case. If that isn't true, then you'd need a clear sentence for the opposite case anyway.

Commented [LP29]: I think you mean 'cf.', but use "compare the" instead

Commented [LP30]: A lot of the Discussion is actually Introduction/Background, that should have told you what to expect before you got to the end. Almost every sentence with cites could have come earlier. As just one example, discussion of the Skarohamar et al. (2015) *model* result could have been used in the Introduction to point to 3-D model support for the more idealized analyses of, e.g., Middleton et al. 1987.

So, thin out the Discussion and strengthen the Introduction.

direction of rotation changes at the slope, and the strength of the
currents increases towards the shelf break, as expected.

The dispersion relation obtained from the model when using bathymetry and
stratification representative for the region (Fig.~\ref{fig:dispcurve})
suggests that diurnal CTWs may be near-resonant, i.e.\ that the group
velocity is zero or close to zero so that energy cannot propagate out of
the area resulting in amplified diurnal tidal currents. A similar result
was obtained by \cite{Middleton1987} using a barotropic shelf wave model
\citep{Saint-Guily1976}. Tidal currents may be enhanced for a range of
frequencies surrounding the RF, and thus RF does not need to coincide
exactly with a tidal frequency for amplification to occur
\citep{Chapman1989}.

The dispersion curves in Fig.~\ref{fig:dispcurve} show that while the
CTWs are relatively barotropic in the region  \citep[according to the
Burger number, $Bu=  \left( \frac{NH}{fL}\right) ^2$, $Bu\ll1$, where
$N^2$ is stratification, $f$ the Coriolis factor and $H$ and $L$ are
representative depth and length scales,
respectively;][]{Wang1976,Brink2006,Jensen2013}, the dispersion curve is
sensitive to relatively small changes in the stratification.

%%%%%%%%%%%%%%%%%%%%
Time series of the KE associated with the diurnal tides and of the tidal
amplitudes derived by harmonic analysis show that diurnal tidal currents
consistently are enhanced by 30--180\,\% during austral summer.
%%%

Commented [LP31]: 1) How does a dispersion curve
show that the CTWs are relatively barotropic?  I *think* the
answer might be that Mode-1 is the only CTW Mode that
can get close to diurnal frequencies, but the modal
structure refers to the number of zero crossing across-slope,
right? Not the vertical structure.

You then present a Burger number argument, which makes
more sense, but is not explicitly related to the CTW
dispersion curves. Better to just argue based on Burger
number, or extract something from the Brink model that
explicitly demonstrates that the mode(s) is(are) barotropic.

%The diurnal tidal forcing does not vary on seasonal scales and thus
cannot explain the enhancement.

The astronomical diurnal tidal forcing and its seasonal variability
cannot explain the asymmetry of the maxima in diurnal tidal KE.
%%%
Studying a small subset of the moorings, \cite{Foldvik85_bottom_currents}
hypothesised that changing stratification causes the breakdown of the
diurnal tidal currents observed during austral winter of 1968, while
\cite{Foldvik1990} suggested that the seasonality was linked to the
variability of the slope current. Other potential explanations are
seasonal changes in the sea ice cover, as high sea ice concentrations
would potentially dampen the CTWs \citep{Ono2008}, or in wind forcing,
which could potentially excite CTWs \citep{Gordon1987} at diurnal
frequencies. These influences will be discussed below.
%Time series of the KE associated with the tides and of the tidal
amplitudes derived by harmonic analysis show that while the tidal forcing
is constant, the tidal currents consistently are enhanced by 30--180\,\%
during austral summer. This was noted in a subset of the moorings by
\cite{Foldvik1990}, who suggested that the seasonality was linked to the
variability of the slope current. In an earlier study, the breakdown of
the diurnal tidal currents observed during austral winter of 1968 was
hypothesised to result from changing stratification \citep{Foldvik1974}.
Other potential explanations are seasonal changes in the sea ice cover,
as high sea ice concentrations would potentially dampen the CTWs
\citep{Ono2008}, or in wind forcing, that could potentially excite CTWs
\citep{Gordon1987} at diurnal frequencies. These influences will be
discussed below.

**Commented [LP32]:** This is too vague. Be explicit about the magnitude of the "seasonal" variability you would expect from astronomical forcing (basically, what modulation do you expect from the K1/P1 couplet?) and state that it is semi-annual (Figure 5a) not annual (Figure 5b)

**Commented [LP33]:** Mack et al., 2013, GRL, show nicely that sea ice responds to the tides (for the Ross Sea). Padman and Kottmeier, 2000, JGR (already cited) show tidal motion of ice for the Weddell Sea. If ice is mobile, it can't dissipate tidal energy; it is "free drift". It's ice mechanics at high ice concentration that provides the friction needed to reduce tides.

%%%%%%%%%%%%%%%%%%%%

[revised manuscript text omitted]
 \citep[Fig.~\ref{fig:pycnovary}a,][]{Jensen2013}. The existence and strength of diurnal \citep[and longer period, ][]{Jensen2013} CTWs in the region must hence be expected to directly influence the availability of warm water above the shelf depth, i.e.\ at depths where it can potentially access the continental shelf. Furthermore, it was recently shown that elevated turbulence levels in the shelf-break region are linked to the semi-diurnal tide and the co-location of critical slope and critical latitude \citep{Fer2016}. The tides thus influence the water mass properties (through mixing) as well as the strength and depth of the thermocline at the shelf break. Modelling efforts aiming to describe and predict the oceanic heat transport towards the FRIS cavity inflow thus

**Commented [LP37]:** Not the right cite. Various available, including Kowalik and Proshutinsky 1994, Padman and Kottmeier, 2000; Mack et al., 2013.

**Commented [LP38]:** I'm not sure why this figure only comes up in Discussion: seems like "Results" to me.

**Commented [LP39]:** Raising warm water above the shelf-break depth does not, by itself, do anything. *Ignoring friction*, CTWs just conserve vorticity; "the tide goes up, the tide goes down".

So, the key to this being important lies in coupling, either with mean flows or friction. Or, as some people have studied, the rectified flows that arise from tidal interactions over sloping topography.

Your paper is not about these processes, and doesn't need to be, but this section needs a little more information to avoid being misleading.

ought to include tidal forcing to correctly capture the dynamics at the shelf break.

Finally, we mention that the observed diurnal tidal currents are up to one order of magnitude larger than those predicted by the tidal model CATS2008 \citep[Fig.~\ref{fig:allplot},][]{Padman2002}, and that the predictions from CATS do not reveal the summer enhancement.
Moreover, due to errors in the model bathymetry, the predicted peak tidal currents are not consistently aligned with the shelf break when running CATS along a cross-shelf section through the locations of moorings M1 and M2 (see Fig.~\ref{fig:allplot} and Fig.~\ref{fig:map} for location of section).
Hence, care must be taken when using CATS to de-tide velocity observations from the region.

%%%%%%%%%%%%%%%%%%%%%%%%%%%%%%%%%%%%%%%%%%%%%%%

\conclusions%[alternative title]

Velocity measurements at 29 moorings located on the continental slope and shelf in the southern Weddell Sea from the period 1968 to 2014 show pronounced diurnal tidal variability. Diurnal tidal currents are strongest at the shelf break and substantially enhanced during austral summer. The summer enhancement is not predicted by the tidal model CATS2008 \citep{Padman2002}.
We investigated the possibility for near-resonant CTWs causing the enhanced diurnal tidal currents by using a 2\,D-numerical code to obtain

**Commented [LP40]:** Yes? Even if not, specify exactly which one.

**Commented [LP41]:** This is a bit unfair: CATS, like every other tide model, is a barotropic model with only tides included.

More honest to say

"… CATS2008 (Figure 11), which does not include stratification or the variability of mean circulation required to predict seasonal modulation of tidal current."

**Commented [LP42]:** Repeating preceding comment:

This is a bit unfair: CATS, like every other tide model, is a barotropic model with only tides included.

More honest to say

"… CATS2008 (Figure 11), which does not include stratification or the variability of mean circulation required to predict seasonal modulation of tidal current."

CTW properties \citep{Brink2006}. Dispersion curves of mode 1 CTWs have a maximum in frequency (the resonant frequency, or "RF"), which results in zero group velocity, i.e.\ trapped energy. The RF moves in and out of the diurnal tidal frequency band depending on the stratification and the slope current which both vary seasonally as hydrographic and current observations at the shelf break reveal. For the weakly stratified water column and strong slope current during austral winter, the RF is found above the diurnal band, suggesting the generation of weak, non-resonant tidal CTWs which quickly propagate out of the generation area. For austral summer conditions, i.e.\ a more stratified upper water column combined with a weaker slope current, the RF can fall into the diurnal band, thus leading to near-resonant diurnal CTWs enhancing the tidal currents.

While no direct influence of wind on the diurnal tidal currents has been found, the varying bathymetry east of the study area as well as the sea ice cover likely affect the propagation of the CTWs. Further studies, using 3\,D-models for example, are needed to quantify these influences as well as to detect the generation site of the CTWs.

The shelf break region in the southern Weddell Sea is an area of great climatic interest. Cold, dense Ice Shelf Water descends the continental slope and contributes eventually to the formation of Antarctic Bottom Water, while warm MWDW flowing onto the shelf prospectively may reach the cavity below FRIS, thus enhancing basal melt rates. The strong diurnal tidal currents at the shelf break facilitate the cross-shelf exchange of water masses and contribute to mixing, hence influencing the hydrographic properties of both the cold outflow and warm inflow.

**Commented [LP43]:** In Conclusions, minimize acronyms even if explained earlier. If useful here, explain them again.

**Commented [LP44]:** Huh?! How can you say "likely" when you have no direct evidence of something? Maybe the problem here is mixing the diurnal wind forcing issue with the others. I think your conclusions are:

"No evidence for wind forcing of diurnal tidal currents."

"No evidence that sea ice affects the diurnal CTWs."

"Varying bathymetry east of the study area likely affects the CTWs seen in the study area."

However, varying bathymetry anywhere affects the CTWs, so why only comment here on the upstream?

%The shelf break area in the southern Weddell Sea is a region of high climatic interest - cold, dense ISW from the FRIS cavity descends the continental slope and contribute to bottom water formation \citep{Foldvik2004} and future scenarios suggest that the inflow of WDW towards the cavity will greatly enhance FRIS basal melt rates in the near future \citep{Hellmer2012}. The strong tidal currents will influence the cross shelf break transport and the hydrographic properties (through mixing) of both the cold outflow and the warm inflow.
% oNE LAST SENTENCE ON THE NEED FOR A 3-d MODEL A LA sKARDHAMAR?

%\appendix
%\section{}    %% Appendix A
%\subsection{}  %% Appendix A1, A2, etc.

\begin{acknowledgements}
For deployment and recovery of moorings, we would like to thank AWI and the crew and scientists on RV \textit{Polarstern} cruises PS08 (recovery of moorings D1, D2, S2-1985 and S3), PS12 (deployment S2-1987), PS34 (deployment Fr1 and Fr2), PS53 (recovery F1--4) and PS82 (recovery SB, SC, SD and SE).
We would also like to thank K. Brink for sharing the numerical code and I. Fer for helpful comments and suggestions.
The research was partially funded by the Centre for Climate Dynamics at the Bjerknes Centre.
\end{acknowledgements}

---

## Referee Comment (RC3) · Anonymous Referee #3 · 2 Aug 2016

This paper deals with Diurnal Topographic Waves (DTW) in the Weddell Sea. The paper is motivated by mooring data collected over several decades and these are discussed on the basis of results from an idealized code of Coastally Trapped Waves. In general the paper is well written and should be suited for publication. There are however some points that the authors could consider prior to publication.

For completeness I miss a figure of the mean velocity based on the moorings, could this be added onto on of the figures? This is of relevance for the choice of current profiles across the shelf break, and further a seasonal variability of the current velocities could be discussed in terms of DTWs summer to winter difference.

Specific comments:

[Figure]

Line 12: Use "weak stratification" instead of "low stratification"

Line 72: Change to "(Brink, 2006) to investigate . . ."

312-323: The authors conclude that the summer amplification of the DTWs during austral summer is not explained by wind. Is there a possibility that the opposite could be the case, i.e. that the increased storminess has a destructive effect on the DTWs during the austral winter? Line 54-55: Is there a reference to accompany this sentence?

Fig 1 and Fig 10: There might be some confusion about what is the positive along-slope current direction; in Fig 1 this would be toward the east while in Fig 10 this appear to be toward the west. Any particular reason why not having positive values with the coast to the left in all cases? Line 80-85: More information is needed here, e.g. the calculation of the orientation of beta requires some choice of averaging length scale that needs some motivation.

Figure 4. An alternative way of including the rotational properties of the velocity series could be to plot them as rotational ellipses (major and minor axis), together with as already done different colors for CW and ACW.

Line 167: In most instances the authors use "austral " winter/summer. Not always . For clarity be sure that this is consistent through the ms.

Line 186-187: "The bathymetry represents an average of six across-slope sections in the area of moorings M1 to M5". I understand that it is necessary to make some representative bathymetry, but some more details and motivation would be good.

Line 199. What is meant by the M-mooring array?

Figure 8. Legend is not easy to read. In particular I have problem with what is meant by the "ref top" and ref 80. Please clarify this.

Line 247: Maybe change to ".. a 40 km wide current with a westward core velocity of 0.2 m s-1 and . . . ".

[Figure]

Line 263: Change "to coincide with a tidal" to "to coincide with one exact tidal .."

Fig 5. This figure must be improved and better simplified.

Line 356-358: Somewhat unclear what is meant here. Do you suggest that the semi-diurnal tide is what sets the stratification that provide the conditions for DTW, or is the point that semi-diurnal is major cause for mixing at the shelf break. Since this point is already mentioned in the introduction a possibility is simply to delete it here.

––––––––––––––––––––––

---

## Author Comment (AC1) · 5 Oct 2016

**Authors' response**

We thank the three reviewers for the comments and appreciate their suggestions which improved the manuscript. In the following, the comments from the referees are included (in black) as well as our response (in blue). Additionally, we provide a marked-up version of the manuscript.

**Reviewer #1**

Review of "Seasonal resonance of diurnal coastal trapped waves in the southern Weddell Sea, Antarctica" By S. Semper and E. Darelius

The authors provide observational evidence for large amplitude diurnal tides on the Weddell Sea slope, and then they seek to rationalize this finding in terms of resonant (zero group velocity) coastal-trapped waves. They go on to demonstrate the conditions (notably mean flow and stratification) that can make the resonant frequency vary from time to time. Overall, this is a credible piece of work, although there are perhaps some places where a bit more could be done.

The main extension I see would be to look at records from the same (or nearly so) isobath, but separated alongshore. These pairs can then be used to estimate alongshore wavelengths. This makes the most sense when moorings are simultaneous, but even if they are not, differences in Greenwich phase could be used to see if the estimated wavelength is at least the right magnitude. Also, the model results provide other information, such as direction of current vector rotation and amplitude of current components, that could also be used to compare with observations. Maybe all we will learn is that things have the right magnitude, but it could be more rewarding.

Reply: We included calculations of the wavelength for the two pairs of simultaneous moorings from nearly similar isobaths in the data set: M1–M4 and M2–M5. They suggest wavelengths of 250–600 km, with a larger range of 200–1600 km for $O_1$ during austral summer. This is roughly consistent with the wavelengths obtained for tidal frequencies in model runs that allow for tidal CTWs, although the RF wavelength typically is higher ($>1000$ km).

The direction of current vector rotation from the numerical code can be seen in Fig. 12 (background colours); the direction changes on the continental slope and the rotational direction is consistent with what the observations suggest. The amplitude provided by the numerical code is arbitrary and can therefore not directly be compared to the observations.

I recall reading somewhere that a PhD thesis by M. Spillane (Oregon State, 1980) showed that the wave dispersion curves can do strange things when inviscid group velocity vanishes. However, I do not know of anything quite like this is the normal literature. It would be interesting to know more about this but it would be asking too much to have this effect covered in this submission.

Reply: We had a look at the PhD thesis you mentioned. You are right, there is a figure of dispersion relations for viscous-damped free shelf waves for different Ekman depths where a singularity occurs at the resonant frequency. However, we have not seen this behaviour being reported in any other publication, and we agree that a more thorough investigation of this effect is beyond the scope of the work here.

Some specific points

- Line 37: Given the mooring notation, it is particularly important here to use the correct (subscripted) tidal notation consistently, e.g., is this M_2 or M2? (Yes, it IS clearly done right in other places).

Reply: Indeed, here the tidal constituent $M_2$ and not mooring M2 is meant. The typo has been corrected.

- 95: Please say more about the CATS tidal model. Is it barotropic? Does it include non-tidal currents? Is it nonlinear? Etc.

Reply: You are right, here is more information needed. We explained the CATS model more detailed now (see also the comment LP12 in the supplement from review #2): "Diurnal tidal KE has also been inferred using tidal predictions from the Circum-Antarctic Tidal Simulation version 2008b (CATS2008b), an updated version of the linear tidal inverse model described by Padman et al. (2002). The barotropic currents at the specific tidal frequencies are predicted for the respective time and location of every mooring deployment and are treated in the same way as the observational current velocities."

- 121: Say here how thick the boundary layer is.

Reply: The boundary layer is more than 25 m thick, as we see its effects at the lowest measurement level at 25 m.a.b., but not at the instrument level above which varies from mooring to mooring (55 m.a.b. for mooring M5, 442 m.a.b. for mooring M4, see Table 1. The text has now been changed to: "The energy associated with the diurnal tidal currents, the diurnal tidal KE (Sect. 2), shows little variation with depth, except at the

lowest measurement level at 25 m.a.b. In this bottom boundary layer, the diurnal tidal KE is slightly decreased compared to the overlying water column (Fig. 4). Depth-averaged diurnal tidal KE is used for further analysis."

- Page 5: Why normalize tidal KE? You are throwing out useful information (actual amplitude), and I do not see any advantage for normalizing.
Reply: In Figure 5a (now 6a), we normalised tidal KE to illustrate and compare its seasonal pattern better, i.e. the asymmetry between summer and winter maxima. The actual amplitudes are given in parentheses in the legend; the summer maximum varies from $10\,\mathrm{cm^2\,s^{-2}}$ at mooring M5 to $203\,\mathrm{cm^2\,s^{-2}}$ at mooring M3. Because of this large spread (resulting from the different locations of the moorings on the deeper slope, the shelf or at the shelf break), we decided to normalise tidal KE.

- 170: The evidence here on ambient currents vis a vis waves is pretty weak. True, you can say that the "mean" alongshore currents vary with time, but, evidently useful local information about the time dependence and the spatial structure is lacking. There is nothing you can do about this, of course, but it would be well to advise the reader that the main thing you can glean here is the magnitude of the "mean" flow, and that it (probably) does vary from time to time.
Reply: In this paragraph, we review what is currently known about the slope current and its seasonal variability upstream, and extend the description with information from our moorings. We have now a) included mean currents in Fig. 1, b) provided information about seasonal variability in the text and c) added a final statement summarising the results that can be used in the modelling: "The limited observations available from our study region suggest a westward flowing jet, which is relatively narrow and appears to be centred at the shelf break. The jet intensifies and widens during early austral winter."

- 243-244: This sentence does not make sense to me.
Reply: We compare the sensitivity of the RF for a change in current core location to a change in current core velocity, since the latter is shown in Fig. 10. We rewrote the paragraph for clarification: "The magnitude of the change in RF for a 40 km off-shore shift depends on the width of the current, but it is comparable to a change in current velocity of $\pm 10\,\mathrm{cm\,s^{-1}}$ (Fig. 11)."

- 264: I am not sure what is meant by a dispersion curve showing the waves to be barotropic. There must be a missing step in the argument here. Maybe they mean the

modal structure (Fig 11) is barotropic? Also, give a representative range of Bu. It is probably a wording issue, but the last sentence of this paragraph seems to contradict the preceding text.

Reply: Yes, the modal structure (Fig. 12) and the observed values of tidal KE (Fig 4) show barotropic motion – as suggested by the low Burger number. We now provide an estimate of $Bu$ and the quantities used to calculated it. Results from the numerical code show however, that while the motion is close to barotropic, the dispersion curve is sensitive to relatively small changes in stratification (Fig. 9-10).

- 270: Enhanced relative to what?

Reply: We added "compared to austral winter" to the sentence to make it clearer.

- 272: What asymmetry is that?

Reply: We meant the difference in amplitude of the summer and winter maximum (see Fig. 6a). The text has been rewritten.

- Figure 7: Label axes.

Reply: It is unclear which axes the reviewer refers to. All axes are labelled in Fig. 7a (now 8a), while Fig. 7b (now 8b) is a schematic illustration of the parameters changed in the sensitivity test and has therefore no scale or units for the axes.

Again, I believe that this contribution is sound overall, but that improvements ought to be made.

Thank you!

**Reviewer #2**

This paper is a comprehensive assessment of the seasonal variability of diurnal tidal currents in the southern Weddell Sea, using data from 29 moorings between 1968 and 2014. While the seasonal variability is known, and has been previously interpreted as response of diurnal tide-forced shelf waves to changing stratification and along-slope current (various papers from the 1980's), the increased data base and interpretation through sensitivity tests on an idealized model (Brink, 2006) makes this a valuable new study.

My specific comments are provided as margin comments and edits on the marked *.tex file. Most of these are relatively minor. Some major comments are as follows:

1. Lines 43-55, about CTWs, will possibly make no sense to a lot of readers without a dispersion curve to look at. Probably this means adding a simple sketch of one, showing that cp is always with shallow water on the left (in the Southern Hemisphere), then three cases of cg (+ve, -ve, and 0 (RF)).
Reply: We have included a schematic as suggested to complement the text.

2. Much of the Discussion is actually Introduction (Background) material. Move everything you knew, or should have known, before the study into Introduction. Discussion could keep some implications (regarding sea ice, mixing etc) that depend on the magnitude of the results you have presented, but expectations should be set in Introduction.
Reply: We have moved the background material from the discussion to the introduction as suggested. This involved some rearrangements in the discussion section.

3. The discussion of Figure 5 is not very clear. I think the argument you are trying to make is that the "diurnal band" as a whole shows mainly semi-annual, which you ascribe to K1/P1 modulation. Then, K1 (removing P1 influence by inference) is "annual". But the modulation of the diurnal band at the semi-annual frequency is too large for K1+P1, even without the other tidal lines (e.g., O1) being caught in the definition of "diurnal band". Some more thought about this, and an improved discussion, would be useful. It is not impossible that currents and stratification add to a semi-annual term in CTW properties.
Reply: We have discussed this point with L. Padman, and the comment is partly due to a mis-reading of Fig 5a (now 6a), which shows KE, and not tidal amplitude as L. Padman first thought. To show that the K1+P1 interference can explain the semi-annual signal we observe, we have included time series of KE obtained from CATS2008b (Fig. 6b). The observed semi-annual modulation of about 0.3-1 is in the upper range of what can be expected and can to a large extent be explained by $K_1+P_1$ modulation. We now discuss other potential contributors – as suggested by L. Padman – to the semi-annual modulation in the discussion.

4) What does it mean to have a wave whose wavelength ($\sim$1300 km) is an order of magnitude longer than typical along-slope scales of isobath variability?
Reply: While this is an interesting question, it is beyond the scope of the current study. We now mention this in the ms, and note that the CTWs modelled by Skardhamar et al. (2015) similarly have wavelengths which are considerably larger than the topographic

scales. A full 3 D-analysis, similar to the one by Skardhamar et al. (2015), would be needed to fully explore the effect of bathymetry on CTWs in the study region.

– Laurie Padman

Please also note the supplement to this comment: http://www.ocean-sci-discuss.net/os-2016-36/os-2016-36-RC2-supplement.pdf

**Supplement**

Thank you for your thorough reading and the comments. The replies to the supplementary comments are given below and are kept rather short, as the comments are.

LP1: The tide-forced CTW's *are* tides; so maybe better here to say "Near-resonance of diurnal tidal CTWs during ..."
Reply: You are right, it might have been confusing. We followed your suggestion and rewrote the sentence.

LP2: I think the Introduction needs to include Nicholls et al., 2009 Reviews of Geophysics. Also, a lot of Discussion should really be moved to Introduction. Anything relevant that you knew or should have known before starting the study should be in the Introduction.
Reply: We now refer to Nicholls et al. (2009) as suggested, and we have moved parts of the discussion into the introduction (see also your comment 2 in the main review).

LP3: Seems to ignore the HSSW contribution to AABW formation.
Reply: While HSSW descends the slope and contributes to AABW formation farther west in the Weddell Sea, ISW is the main contributor in our study region. But ISW is again HSSW+meltwater, so including both water masses (while keeping the text short), we now write: "This is where cold and dense water masses, formed on the continental shelf and underneath the Filchner-Ronne Ice Shelf (FRIS), cross the shelf break and descend the continental slope (Foster and Carmack, 1976; Foldvik et al., 2004; Nicholls et al., 2009).

LP4: Strange sentence. What about the inflow and outflow are "influenced", vs "to some extent set" ? And how do these two things even differ?
Reply: There is no real difference, so we reworded the sentence: "Physical processes at the shelf break and on the continental slope influence both the cold outflow and the warm

inflow in terms of their hydrographic properties and strengths."

LP5: Note sure I got the TeX right, but just include Darelius as a regular cite.
Reply: It is a TeX thing (our key name for this article).

LP6: You need to explain the source of the mixing better here. I think you are referring to bottom stress. If you want to claim a baroclinic source of mixing, then needs a cite. Fer et al. paper in prep. would be good, but maybe Ilker's Yermak Plateau paper?
Reply: Yes, this was unclear. We included two references and rewrote the sentence: "Mixing can be expected to be further enhanced at the shelf break by the strong diurnal tidal currents (Fer et al., 2015; Pereira et al, 2002)."

LP7: I think this discussion will not make much sense to most readers until you show a dispersion curve for a CTW, even if it is just a schematic showing cp (always shallow water on left), and the three cases of cg>0, cg<0, and RF.
Reply: See your comment 1 in the main review.

LP8: This expression isn't clear; probably need a little introduction to bathymetric irregularity, convergence etc.
Reply: This is a good point; the reader should get a slightly more detailed explanation. We rephrased the sentence: "In practice, energy likely escapes in one or the other direction along the slope. Leakage of energy occurs for example because of irregularities in the bathymetry and because the bottom slope changes (i.e. isobaths converge or diverge, Thomson and Crawford, 1982). Therefore, we use the term near-resonance rather than resonance."

LP9: This approach is sensible, but relies on choosing a length scale for the calculation of local isobaths orientation.
Reply: See comment by reviewer #3.

LP10: Better word than "chunk" ? Maybe "intervals" ?
Reply: Agree, "intervals" is a better word. We changed the paragraph accordingly.

LP11: So, how long is each window? I think this needs 3/4 of a month to work, but that is a bad window length for tides. Better to use 14 or 29 days (more precisely, the spring/neap for O1/K1)

Reply: Sorry for the confusion. The window length we used is 14 days (a third of 1.5 months), but since they are overlapping, there is actually five windows within each interval. We corrected the paragraph: "The hourly-averaged current meter data are divided into intervals of 1.5 months length beginning every 14th day. For each interval, the power spectral densities are estimated using Welch's method (Welch,1976) and 14 day long, 50 %-overlapping Hanning windows."

LP12: Need to be clear about which version of CATS you are using. The cite you give is for CATS, or CADA00.10, both very old. I think you use CATS2008. If so, the cite would be
"... Simulation version 2008b (CATS2008b) an updated version of the tidal inverse model described by Padman et al. (2002)."
Reply: Yes, we are using CATS2008b, but did not know how to cite it properly. Thanks for correcting this!

LP13: This is probably good, but it seems a little bit strange since the energy in the tide model is known exactly for exact frequencies.
Reply: We are aware of this, but decided to treat both observational and model data in a consistent way.

LP14: Mike Foreman did the really hard work here!
Reply: Thanks for pointing this out, we were not aware of this! We followed your suggestion and cited his work.

LP15: 1) I guess you don't do much with the semidiurnals, but you'd also need to use inference to separate S2 and K2.
2) Inference on K1/P1 assumes that the relative amplitude and phases are constant throughout the year. Given sensitivity of actual CTW currents to exact frequency, is this valid?
Reply: 1) Correct, we do not do anything with the semidiurnal tides, so we did not see the need for separating S2 and K2. 2) We cannot claim that the relative amplitude and phases are constant throughout the year. Hence, we removed the previous Fig. 5b where we used the separation of $K_1$ and $P_1$ and also removed the sentence in the methods you refer to here.

LP16: Topographic setting is the key to CTWs, right? Not the outflow.

Reply: Right. We changed the sentence following your suggestion.

LP17: Figure 3 caption needs to tell us the water depth for each of M5 and M4.
Reply: Thanks, that is a good idea. We changed the caption to "Diurnal tidal KE over time and depth at a) mooring M5 and b) mooring M4 on the continental slope, located above the 1900 m and 1050 m isobath, respectively."

LP18: This figure needs two panels, one for summer and the other for winter.
Reply: We do have a similar figure for austral winter, but decided to not show it. Since the area of the circles is proportional to the square root of the maximum of depth-averaged, diurnal tidal KE, the summer–winter differences are not easy to see. Here we rather want to show the spatial distribution of maximum tidal KE.

LP19: You need to talk about what it looks like *west* of the trough, which is different from east of the trough. Otherwise, it looks like you are hiding something. (Are you?!)
Reply: No, we are not trying to hide anything. At moorings A and B2, the major axes are directed mostly across-slope, similar to the eastern part of the study area. For the F-, D-, and W-moorings, the picture is more complicated. They are located close to the ridge which might influence both the accuracy of the chosen rotation angle and the tidal currents over this bathymetric barrier. We rewrote the sentence: "The major axes of the tidal ellipses at the $K_1$ frequency are directed across the continental slope for moorings located at the shelf break and on the continental slope (especially for the ones east of the Filchner Depression and west of the ridge). Hence, the diurnal tidal energy is higher in the across-slope component than in the along-slope component."

LP20: Something is wrong here! P1 and K1 don't explain this high degree of semiannual variability, especially as the "diurnal band" still contains O1.
Reply: As explained in comment 3 in the main document, this comment is due to a misreading of the figure labels. Figure 5a (now 6a) shows diurnal tidal KE, not tidal amplitude. The semi-annual modulation by $K_1+P_1$ can to a large degree explain the observed signal in diurnal tidal KE.

LP21: This is a nice figure. However, is it really a "Hovmoller" diagram as stated in the caption? `https://en.wikipedia.org/wiki/Hovm%C3%B6ller_diagram`
Reply: Thanks. Yes, it is a Hovmöller diagram. Following your link, it says: "Hovmöller diagrams are also used to plot the time evolution of vertical profiles of scalar quantities such as temperature, density, or concentrations of constituents in the atmosphere or ocean. In that case time is plotted along the abscissa and vertical position (depth, height, pressure) along the ordinate."

LP22: I disagree: In your Figure 6, if I plotted T and S at the bottom of the plot, I would see a seasonal signal.
Reply: At mooring M3 shown in the figure (now Fig. 7), it is the thermocline that varies in depth over the year. It reaches the bottom during austral winter, which leads to the apparent seasonal signal you refer to. However, at moorings located farther down the continental slope (M5, for example), no seasonal signal is present below the thermocline. We changed the sentence to avoid confusion about this: "The deeper moorings (e.g. M5) show that seasonal changes in the water column below the thermocline are negligible (not shown)."

LP23: You don't really have a "level of the code". Really, all you need here is
Reply: This comment is incomplete, but we have removed the "level of the code".

LP24: If the wavelength is really this long, does it make sense to think of these as 'waves' when along- slope topography varies on much smaller scales?
i.e., Discuss implications for wavelength >> topo scales
Reply: See your comment 4 in the main review. We can not address this point with the tools at hand, but we note that the wavelengths of the tidal CTWs modelled by Skardhamar et al. (2015) also are larger that the scale of along slope variability.

LP25: It's been so long since you mentioned this, you might need to explain it again.
Reply: We are not in favour of re-defining abbreviations, but see your point and changed to: "For tidal CTWs to exist, the dispersion curve must pass through the tidal band, i.e. the maximum of the dispersion curve (the resonant frequency, or "RF") must lie within (thus giving near-resonance) or above the diurnal tidal frequency band."

LP26: So ... why not use higher resolution in the Brink model? This is the obvious thing to do. I know there are some stability issues with this code, and maybe that's the answer. But if this is the reason, you need to discuss it here, or maybe better in Section 4.1.
Reply: You are right, it is the stability issue that prevented us from simply increasing the resolution. We agree that we should discuss this, and did so in Section 4.1: "The code was set up using 30 vertical levels and 120 horizontal grid points to represent a 2 D-cross-slope

section. This is within the recommended range of grid points; increasing the resolution leads to instability and failure of the test for hydrostatic consistency."

LP27: Figure 10 needs O1 and K1 frequency lines marked.
Reply: This is a good idea. We included the $O_1$ frequency line only, as $K_1$ lies much higher (note the small scale of the y-axis).

LP28: I do not like this bracketed way of doing opposite cases. In general, you can ignore the bracketed examples as they are implied as the opposite of the main case. If that isn't true, then you'd need a clear sentence for the opposite case anyway.
Reply: Changed, see comment by reviewer #1.

LP29: I think you mean 'cf.', but use "compare the" instead
Reply: Yes, we meant cf. and changed it accordingly.

LP30: A lot of the Discussion is actually Introduction/Background, that should have told you what to expect before you got to the end. Almost every sentence with cites could have come earlier. As just one example, discussion of the Skarohamar et al. (2015) *model* result could have been used in the Introduction to point to 3-D model support for the more idealized analyses of, e.g., Middleton et al. 1987.
So, thin out the Discussion and strengthen the Introduction.
Reply: See your comment 2 in the main review.

LP31: 1) How does a dispersion curve show that the CTWs are relatively barotropic? I *think* the answer might be that Mode-1 is the only CTW Mode that can get close to diurnal frequencies, but the modal structure refers to the number of zero crossing across-slope, right? Not the vertical structure.
2) You then present a Burger number argument, which makes more sense, but is not explicitly related to the CTW dispersion curves. Better to just argue based on Burger number, or extract something from the Brink model that explicitly demonstrates that the mode(s) is(are) barotropic.
Reply: This was worded confusingly. See also comment by reviewer #1. The modal structure (Fig. 12) and the observed values of tidal KE (Fig 4) show barotropic motion – as suggested by the low Burger number. We now provide an estimate of $Bu$ and the quantities used to calculated it. Results from the numerical code show however, that while the motion is close to barotropic, the dispersion curve is sensitive to relatively small

changes in stratification (Fig. 9-10).

LP32: This is too vague. Be explicit about the magnitude of the "seasonal" variability you would expect from astronomical forcing (basically, what modulation do you expect from the K1/P1 couplet?) and state that it is semi-annual (Figure 5a) not annual (Figure 5b)
Reply: We now included a more comprehensive paragraph on the $P_1$-$K_1$-coupling (Fig. 5a, now 6a,b) and removed the previous Fig. 5b.

LP33: Mack et al., 2013, GRL, show nicely that sea ice responds to the tides (for the Ross Sea). Padman and Kottmeier, 2000, JGR (already cited) show tidal motion of ice for the Weddell Sea. If ice is mobile, it can't dissipate tidal energy; it is "free drift". It's ice mechanics at high ice concentration that provides the friction needed to reduce tides.
Reply: We included your information in the text.

LP34-36: All Introduction text
Reply: We rewrote the introduction and discussion, as you suggested in comment 2 in the main review.

LP37: Not the right cite. Various available, including Kowalik and Proshutinsky 1994, Padman and Kottmeier, 2000; Mack et al., 2013.
Reply: Sorry, you are right, this was a typo. Anyway it is good to include several references, which we did now.

LP38: I'm not sure why this figure only comes up in Discussion: seems like "Results" to me.
Reply: This figure illustrates the consequences or effects the strong diurnal tidal currents and CTWs have. To us, this seems to fit better in the discussion than at any place in the results.

LP39: Raising warm water above the shelf- break depth does not, by itself, do anything. *Ignoring friction*, CTWs just conserve vorticity; "the tide goes up, the tide goes down". So, the key to this being important lies in coupling, either with mean flows or friction. Or, as some people have studied, the rectified flows that arise from tidal interactions over sloping topography.
Your paper is not about these processes, and doesn't need to be, but this section needs a

little more information to avoid being misleading.

Reply: We agree that more information will help the understanding. We rewrote as: "The existence and strength of diurnal (and longer period, Jensen et al., 2013) CTWs in the region must hence be expected to directly influence the availability of warm water above the shelf depth, i.e. at depths where it can potentially access the continental shelf through the influence of other processes such as e.g. a background mean flow, rectified tidal flows, friction or eddy exchanges."

LP40: Yes? Even if not, specify exactly which one.

Reply: Indeed, we should specify the version of the tidal model CATS, which we did in the revised version of the manuscript (see LP41 & LP42).

LP41 & LP42: This is a bit unfair: CATS, like every other tide model, is a barotropic model with only tides included.
More honest to say
"... CATS2008 (Figure 11), which does not include stratification or the variability of mean circulation required to predict seasonal modulation of tidal current."

Reply: We agree, this was not formulated well, but was not intended as criticism against the model. We rewrote the paragraph following your suggestion.

LP43: In Conclusions, minimize acronyms even if explained earlier. If useful here, explain them again.

Reply: We followed your suggestion as earlier (LP25) and explained the acronym again.

LP44: Huh?! How can you say "likely" when you have no direct evidence of something? Maybe the problem here is mixing the diurnal wind forcing issue with the others. I think your conclusions are:
"No evidence for wind forcing of diurnal tidal currents."
"No evidence that sea ice affects the diurnal CTWs."
"Varying bathymetry east of the study area likely affects the CTWs seen in the study area."
However, varying bathymetry anywhere affects the CTWs, so why only comment here on the upstream?

Reply: You are right, this was confusingly formulated. We comment on the bathymetry in the east as it is much steeper there, i.e. there is a strong divergence of isobaths into the study area, while the differences west of the study area are less pronounced. We changed

the sentence to: "While no direct influence of wind on the diurnal tidal currents and no evidence of sea ice affecting the diurnal CTWs have been found, the varying bathymetry east of the study area likely affects the propagation of the CTWs."

**Reviewer #3**

This paper deals with Diurnal Topographic Waves (DTW) in the Weddell Sea. The paper is motivated by mooring data collected over several decades and these are discussed on the basis of results from an idealized code of Coastally Trapped Waves. In general the paper is well written and should be suited for publication. There are however some points that the authors could consider prior to publication.

For completeness I miss a figure of the mean velocity based on the moorings, could this be added onto on of the figures? This is of relevance for the choice of current profiles across the shelf break, and further a seasonal variability of the current velocities could be discussed in terms of DTWs summer to winter difference.
Reply: This sounds like a good idea. However, most of the moorings are located in the Ice Shelf Water plume, which flows out of the Filchner Depression, and the velocities will be influenced by that (see Fig. 2 in Foldvik et al.,2004). We added the mean velocities of the moorings that are supposedly least influenced by the plume onto the map in Fig. 1 and included more information about current velocities in the results.

Specific comments:

Line 12: Use "weak stratification" instead of "low stratification"
Reply: Thanks for pointing this out. We changed from "weak" to "low" and checked the manuscript for consistency on this.

Line 72: Change to "(Brink, 2006) to investigate . . ."
Reply: Changed according to your suggestion, the sentence reads more easily now.

312-323: The authors conclude that the summer amplification of the DTWs during austral summer is not explained by wind. Is there a possibility that the opposite could be the case, i.e. that the increased storminess has a destructive effect on the DTWs during the austral winter?
Reply: According to Gordon and Huthnance (1987), short-duration storms have been

observed to excite near-resonant mode 1 CTWs. Wind is generally a means of forcing CTWs (Huthnance et al., 1986). We have not found any example for increased storminess having a destructive effect on CTWs, as you suggest. If there is any, we do not consider this effect to be large.

Line 54-55: Is there a reference to accompany this sentence?
Reply: We rewrote the sentence as follows, including a reference (see also comment LP8 in the supplement from reviewer #2): "In practice, energy likely escapes in one or the other direction along the slope. Leakage of energy occurs for example because of irregularities in the bathymetry and because the bottom slope changes (i.e. isobaths converge or diverge, Thomson and Crawford, 1982). Therefore, we use the term near-resonance rather than resonance."

Fig 1 and Fig 10: There might be some confusion about what is the positive along-slope current direction; in Fig 1 this would be toward the east while in Fig 10 this appear to be toward the west. Any particular reason why not having positive values with the coast to the left in all cases?
Reply: You are right, this is confusing. For the model set-up, a right-handed coordinate system is used where $u$ is directed on shelf, hence $v$ is positive eastward. We changed Fig. 10 such that the velocities are negative, to be consistent. However, when we mention westward flow in the text, we use positive amplitudes, as the term "westward" already contains information on the direction.

Line 80-85: More information is needed here, e.g. the calculation of the orientation of beta requires some choice of averaging length scale that needs some motivation.
Reply: The average length scale was chosen to be on the order of $10\,\text{km}$. An exact number is difficult to provide, as a useful length scale also depends on the location of the mooring (i.e. shorter scales for moorings close to the ridge where bathymetry changes more). All angles have been checked visually to insure consistency. The text has been changed to: "The rotation angle $\beta$, positive for clockwise rotation, is listed in Table 1; it is inferred for each mooring from the local bathymetry based on the GEBCO_2014 bathymetry grid (The GEBCO_2014 Grid, version 20150318, `http://www.gebco.net`) and using an average length scale of the order of $10\,\text{km}$. The estimated accuracy of $\beta$ is approximately $\pm 10°$."

Figure 4. An alternative way of including the rotational properties of the velocity series

could be to plot them as rotational ellipses (major and minor axis), together with as already done different colors for CW and ACW.

Reply: Thanks for the suggestion, we are aware of this possibility. In Fig. 4 (now 5), we show the orientation of the major axis and the rotation of the current, but want to show the spatial distribution of the maximum tidal KE as well, which is done easiest using circles scaled to the amount of tidal KE.

Line 167: In most instances the authors use "austral " winter/summer. Not always . For clarity be sure that this is consistent through the ms.

Reply: We agree, this should be done consistently throughout the manuscript. We have now included "austral" everywhere where it might be ambiguous.

Line 186-187: "The bathymetry represents an average of six across-slope sections in the area of moorings M1 to M5". I understand that it is necessary to make some representative bathymetry, but some more details and motivation would be good.

Reply: We used the same bathymetry as Jensen et al. (2013) after having compared it to our larger study area. The paragraph has been rephrased to explain this more clearly: "Following Jensen et al. (2013), we use a closed coastal but open offshore boundary, a free surface and a negligible bottom friction. Furthermore, we apply the same bathymetry as Jensen et al. (2013). It represents an average of six across-slope sections with approximately 20 km separation in the area of moorings M1 to M5 and compares well to sections farther west in our study area (not shown)."

Line 199. What is meant by the M-mooring array?

Reply: The M-mooring array comprises moorings M1 to M5. We changed the sentence to avoid possible confusion: "Figure 8a shows the obtained density and stratification profiles, representative for the shelf break at moorings M1 to M5 in austral summer."

Figure 8. Legend is not easy to read. In particular I have problem with what is meant by the "ref top" and ref 80. Please clarify this.

Reply: We changed the legend (and the figure caption). The legend includes now "bathy east", a run based on the reference stratification with the steeper bathymetry east of the study area, and "surface top" and "surface mean". These are runs for the reference stratification with a differently inferred surface $N^2$ value, as the uppermost stratification value in the numerical code covers the upper 160 m. The run "surface top" uses the uppermost value of the observational stratification profile for the stratification in the code, and the

run "surface mean" uses the average of the upper 80 m of the observed stratification profile.

Line 247: Maybe change to ".. a 40 km wide current with a westward core velocity of 0.2 m s-1 and . . . ".
Reply: Thanks for your suggestion, which we followed. The sentence reads now more easily.

Line 263: Change "to coincide with a tidal" to "to coincide with one exact tidal .."
Reply: We changed the sentence according to your suggestion.

Fig 5. This figure must be improved and better simplified.
Reply: See comments from the other two reviewers. We removed the previous Fig. 5b and explained the contents of panel a) better, with the help of a new panel b).

Line 356-358: Somewhat unclear what is meant here. Do you suggest that the semi-diurnal tide is what sets the stratification that provide the conditions for DTW, or is the point that semi-diurnal is major cause for mixing at the shelf break. Since this point is already mentioned in the introduction a possibility is simply to delete it here.
Reply: The latter is what we mean; the semi-diurnal tide is the major cause for mixing at the shelf break. We rewrote the paragraph and changed also the sentence you refer to: "
[revised manuscript text omitted]

---

## Referee Report (RR1)

Re-review of: *Seasonal resonance of diurnal coastal trapped waves in the southern Weddell Sea, Antarctica*, by S. Semper and E. Darelius.

This paper describes evidence that observed modulation of diurnal tidal currents along the outer continental shelf and slope of the southern Weddell Sea can be explained by sensitivity of the dispersion curve for coastal trapped waves (CTWs) to changes in ocean stratification and the strength of the Antarctic Slope Current (ASC). The revised paper is well written and tells a valuable story. I have only relatively minor additional comments for the authors and editor to consider before publication.

As an aside, however, the authors seem to gloss over the fact that everything here was known, from old studies by Foldvik, Middleton and others. The key features of the new work are (a) more current meters, (b) a fairly convincing demonstration that we should worry more about stratification and less about the ASC, and (c) improved knowledge of upper-ocean stratification in winter thanks to seals. While I wouldn't require revision of the paper to clarify this, the authors might keep this in mind as they make any final tweaks to the text.

Major comments:

24-25 and elsewhere: the authors never define how WDW differs from MWDW. Some people use 0 deg. C as the threshold. By that definition, a lot of deep water in Figures 7 and 13 is WDW, not MWDW.

43-44, 73-75, 90-95: and maybe elsewhere in Section 1: I recommend removing all the sentences that talk about what this paper will do, and put them together at the end of Section 1. This will also make it clearer how your work improves on earlier work.

Minor comments:

28: Change to "Some climate models"; Hellmer et al. 2012 actually evaluated several climate models and chose just one or two for forcing their high-resolution ocean/ice model.

49-50: 'to exist and decay' => "to exist, and their energy decays'

50: 'While CTWs propagate with shallow waters to the left …'. Depends how you define propagation. If energy is going to the *right*, that's the important propagation direction from the point of view of regional energetics. Maybe be specific about it being the direction of phase propagation.

63-65: sentence beginning "In our study …". "strong tidal currents" don't "break" down; they weaken (or perhaps disappear). You might do better with "… tidal currents, and they attributed a weakening of currents in winter to …"

71-72: => "Our study is based on a more extensive data set, with observations of current velocities from 29 moorings being used to quantify the strength of diurnal tidal currents and to describe …"

73-75: This is where scattering "we will do" text throughout the Introduction gets messy. As stated here, you are only looking at stratification. But in fact you also look at currents, so the text in line 82 is actually more useful ("change in the oceanographic background {state}").  Move all "roadmap to the paper" text to the end of Introduction.

77-77: You already told us what we could learn from Foldvik et al. (1990 in lines 67-69.

97-99: You don't really need this. If you want something to end with, summarize for us how this advances what was already known from the earlier studies.

103: "are of 1-2 years" => "are each of 1-2 years"

110, 113: beta is an unfortunate symbol to use for this, as oceanographers frequently associate it with Coriolis ('beta-plane').

170: "distances" => "separations"

175-179: We should know at this point what the practical differences between WDW and MWDW are.

306: I don't think you need quotation marks around 'generated'

315: I would NOT describe "by 30% to 180%" as being "CONSISTENTLY enhanced" !

320-321: "The observed FRACTIONAL difference"

321: "although on the high end when compared with" is too vague.

324-327: Clearer alternative text?  "It is possible that other factors contribute to the observed semi-annual variability. For example, there may be a semi-annual cycle of mixing (and hence stratification) caused by the K1–P1 interference. Alternatively, the observed variability of diurnal KE may be the sum of the effect of an annual cycle in stratification that is phase-shifted relative to the effect of changing background current (L. Padman, personal communication, 9 August 2016)."

340-341: change to '… region occur above the pycnocline, similar to …', then delete "above the pycnocline' on the next line.

352: It seems strange to call the SSM "the manifestation of the transition from …"; it really IS the transition, right?

360: number format for range of L: just quote in km (300-500 km).

361: I feel like I've been told about the Foldvik et al. (1990) result too often!

368-369: "The stronger current observed during austral autumn and winter will however add to the effect of the low winter time stratification and move the RF to higher frequencies." This is quite important, and you didn't really explore it in your model tests. That's okay. But here you make it sound like it is a minor factor, whereas the whole concept of resonance is that things are very sensitive to getting RF close to tidal forcing. Maybe make it sound more possible that the SUM of current and stratification may be critical, even if current by itself is not.

371-372: "While tidal energy cannot be dissipated when ice is drifting freely, high ice concentration provides the friction which is needed to reduce tides (Padman et al., 2002)." This sounds like it is plausible that the large reduction is diurnal currents is plausibly a frictional effect. Really, all Padman et al. (2002) said, in the context of their barotropic model, was that you might want a larger drag coefficient to account for non-free-drift ice (including ice shelves, land-fast ice, and high-concentration drifting ice). Here's an alternative, although you might want to rephrase it: "When sea ice is in free drift (Padman and Kottmeier, 2000) no tidal energy is dissipated at the ocean-ice interface. However, as ice concentration increases and internal ice stresses prevent the ice from responding to local tidal currents, the stress at the ocean-ice interface may be significant compared with friction at the seabed, thus removing tidal energy and reducing tidal currents (Padman et al., 1992)."

382-383: "rendering a considerable effect of sea ice questionable." => "inconsistent with the response if damping by ocean-ice interactions was a significant factor."

406 and 416: structure with 'e.g.' is awkward.

452: Change to "Studies with realistic 3D ocean models are needed to …"

---

## Author Response (AR2)

**Authors' response**

We thank the editor J. Huthnance and reviewer #2 L. Padman for their helpful comments on the manuscript. The comments are included in black, our replies to them are given in blue. Additionally, we provide a marked-up version of the revised manuscript.

**Comments from the editor**

Dear Authors

Thank-you again for your revised manuscript. Here as "promised" are a few editorial comments to incorporate please along with any comments from the review now under way. Line numbers refer to the "clean" version of the revised manuscript.

Yours sincerely

John Huthnance

Lines 165-166 and 312-322. In my opinion the essence of semi-annual period in diurnal tides is contained in lines 312-314. The physics behind it is that there is no forcing for diurnal tides if the sun and moon are both over the equator. In fact it is necessary to have three diurnal constituents to represent the $\sim$ 14-day "spring-neap" period and the semi-annual variation. So I don't think the discussion about K1 and P1 is helpful unless you also bring in O1. K1 and O1 are the largest diurnal constituents and produce the $\sim$ 14-day "spring-neap" period. However, these two alone would give the same magnitude of springs and neaps throughout the year (which is why the third constituent is needed). But this might all be unnecessarily complicated; you might just keep lines 312-314 without going into the constituents. Your choice!

Reply: We have rewritten and simplified these sections so that we do not discuss K1+P1 but only the total astronomical forcing of the diurnal constituents. As a consequence, the K1+P1 lines in Fig 6b are removed.

Line 400 "allow for the generation of diurnal CTWs". Do you mean this or just "enhance diurnal CTWs"? "allow for" is strange anyway; maybe "favour"?

Reply: We changed "allow for" to "favour".

Figure 1 caption last line. "rotation" → "orientation"

Reply: We changed "rotation" to "orientation".

Table 1 row "M3". Something is wrong with the instrument depths. 1993 m is deeper

than the water depth and the ranges overlap which is confusing.

Reply: The comma was at the wrong position yielding wrong numbers. It is fixed now.

Figure 5 caption. "The area of the circles is proportional to the square root of the KE." Please check this. It means that the linear scale is proportional to the square root of the speed, or that the area is proportional to the speed. Linear scale proportional to the speed is more usual - but I can see that you have a wide range of KE values so you might mean what you say!

Reply: We mean what we say, it is a non-linear scale due to the wide range of KE values.

**Comments from reviewer #2**

This paper describes evidence that observed modulation of diurnal tidal currents along the outer continental shelf and slope of the southern Weddell Sea can be explained by sensitivity of the dispersion curve for coastal trapped waves (CTWs) to changes in ocean stratification and the strength of the Antarctic Slope Current (ASC). The revised paper is well written and tells a valuable story. I have only relatively minor additional comments for the authors and editor to consider before publication.

As an aside, however, the authors seem to gloss over the fact that everything here was known, from old studies by Foldvik, Middleton and others. The key features of the new work are (a) more current meters, (b) a fairly convincing demonstration that we should worry more about stratification and less about the ASC, and (c) improved knowledge of upper-ocean stratification in winter thanks to seals. While I wouldn't require revision of the paper to clarify this, the authors might keep this in mind as they make any final tweaks to the text.

Reply: When restructuring the introduction, we tried to make this clearer.

Major comments:

24-25 and elsewhere: the authors never define how WDW differs from MWDW. Some people use 0 deg. C as the threshold. By that definition, a lot of deep water in Figures 7 and 13 is WDW, not MWDW.

Reply: We now clearly define WDW and MWDW in terms of their hydrographic properties (lines 179–180 and line 182), and we have made sure to use the right water mass abbreviation in the right place. The water at depth in Fig. 7 and 13 is indeed WDW.

43-44, 73-75, 90-95: and maybe elsewhere in Section 1: I recommend removing all the sentences that talk about what this paper will do, and put them together at the end of Section 1. This will also make it clearer how your work improves on earlier work.
Reply: We followed your advice and restructured the introduction.

Minor comments:

28: Change to "Some climate models"; Hellmer et al. 2012 actually evaluated several climate models and chose just one or two for forcing their high-resolution ocean/ice model.
Reply: We changed "climate models" to "some climate models".

49-50: 'to exist and decay' → "to exist, and their energy decays'
Reply: We changed the sentence according to your suggestion.

50: 'While CTWs propagate with shallow waters to the left . . . '. Depends how you define propagation. If energy is going to the *right*, that's the important propagation direction from the point of view of regional energetics. Maybe be specific about it being the direction of phase propagation.
Reply: We changed the sentence as follows: "While the direction of phase propagation is with shallow water to the left (in the southern hemisphere), the group velocity $c_g$ of CTWs, and thus the energy associated with the waves, can propagate in either direction."

63-65: sentence beginning "In our study . . . ". "strong tidal currents" don't "break" down"; they weaken (or perhaps disappear). You might do better with ". . . tidal currents, and they attributed a weakening of currents in winter to . . . "
Reply: Foldvik et al. (1985b) mention repeatedly the "breakdown of the diurnal tides". We see your point, though, and changed the wording as you suggested.

71-72: → "Our study is based on a more extensive data set, with observations of current velocities from 29 moorings being used to quantify the strength of diurnal tidal currents and to describe . . . "
Reply: The sentence has been rewritten when restructuring the introduction.

73-75: This is where scattering "we will do" text throughout the Introduction gets messy. As stated here, you are only looking at stratification. But in fact you also look at currents, so the text in line 82 is actually more useful ("change in the oceanographic background

state"). Move all "roadmap to the paper" text to the end of Introduction.
Reply: We restructured the introduction.

77-77: You already told us what we could learn from Foldvik et al. (1990) in lines 67-69.
Reply: We rewrote this paragraph since we moved the previous sentences to the end of the introduction.

97-99: You don't really need this. If you want something to end with, summarize for us how this advances what was already known from the earlier studies.
Reply: The paragraph has been changed when restructuring the introduction.

103: "are of 1-2 years" $\rightarrow$ "are each of 1-2 years"
Reply: Changed.

110, 113: beta is an unfortunate symbol to use for this, as oceanographers frequently associate it with Coriolis ('beta-plane').
Reply: We replaced "beta" with "phi". Since we state that this is the rotation angle for the rotation of the coordinate system, there should not be any confusion.

170: "distances" $\rightarrow$ "separations"
Reply: Changed.

175-179: We should know at this point what the practical differences between WDW and MWDW are.
Reply: See reply to your first major comment.

306: I don't think you need quotation marks around 'generated'
Reply: We deleted the quotation marks.

315: I would NOT describe "by 30% to 180%" as being "CONSISTENTLY enhanced" !
Reply: With "consistently enhanced" we meant that all the moorings show the summer enhancement, not that the magnitude of enhancement is consistent. We rewrote the sentence: "The austral summer peak is enhanced for all moorings by 30 % to 180 % compared to the austral winter peak."

320-321: "The observed FRACTIONAL difference"

Reply: Changed.

321: "although on the high end when compared with" is too vague.
Reply: We now quantify the reduction in CATS compared to the observations. See also first comment from the editor.

324-327: Clearer alternative text? "It is possible that other factors contribute to the observed semi-annual variability. For example, there may be a semi-annual cycle of mixing (and hence stratification) caused by the K1–P1 interference. Alternatively, the observed variability of diurnal KE may be the sum of the effect of an annual cycle in stratification that is phase-shifted relative to the effect of changing background current (L. Padman, personal communication, 9 August 2016)."
Reply: Yes, this is clearer, thanks. Since we do not talk about the K1–P1 interference specifically any longer, we replaced "K1–P1 interference" by "semi-annual tidal signal" in the second sentence.

340-341: change to '... region occur above the pycnocline, similar to ...', then delete "above the pycnocline' on the next line.
Reply: We changed the sentence.

352: It seems strange to call the SSM "the manifestation of the transition from ..."; it really IS the transition, right?
Reply: Yes, it is. We deleted "the manifestation of".

360: number format for range of L: just quote in km (300-500 km).
Reply: OK, changed.

361: I feel like I've been told about the Foldvik et al. (1990) result too often!
Reply: We have removed a few of the references to his work.

368-369: "The stronger current observed during austral autumn and winter will however add to the effect of the low winter time stratification and move the RF to higher frequencies." This is quite important, and you didn't really explore it in your model tests. That's okay. But here you make it sound like it is a minor factor, whereas the whole concept of resonance is that things are very sensitive to getting RF close to tidal forcing. Maybe make it sound more possible that the SUM of current and stratification may be critical,

even if current by itself is not.

Reply: We tried to make the point more explicit by rewriting as: "The combined effect of stratification and current, however, is considerable, as the stronger current observed during austral autumn and winter will add to the effect of the low winter time stratification and move the RF to higher frequencies."

371-372: "While tidal energy cannot be dissipated when ice is drifting freely, high ice concentration provides the friction which is needed to reduce tides (Padman et al., 2002)." This sounds like it is plausible that the large reduction is diurnal currents is plausibly a frictional effect. Really, all Padman et al. (2002) said, in the context of their barotropic model, was that you might want a larger drag coefficient to account for non-free-drift ice (including ice shelves, land-fast ice, and high-concentration drifting ice). Here's an alternative, although you might want to rephrase it: "When sea ice is in free drift (Padman and Kottmeier, 2000) no tidal energy is dissipated at the ocean-ice interface. However, as ice concentration increases and internal ice stresses prevent the ice from responding to local tidal currents, the stress at the ocean-ice interface may be significant compared with friction at the seabed, thus removing tidal energy and reducing tidal currents (Padman et al., 1992)."

Reply: Thanks for this clarification. We changed the paragraph.

382-383: "rendering a considerable effect of sea ice questionable." → "inconsistent with the response if damping by ocean-ice interactions was a significant factor."

Reply: We changed the sentence according to your suggestion.

406 and 416: structure with 'e.g.' is awkward.

Reply: Line 406: We deleted "e.g.": "The wavelengths of the diurnal CTWs are typically large compared to the length scale over which the changes in bathymetry discussed above occur and the scale of other topographic features in the area."

Line 416: We changed "identified e.g." to "here identified": "We note that the depth of the WW–WDW transition (here identified by the -1° C isotherm) ..."

452: Change to "Studies with realistic 3D ocean models are needed to . . ."

Reply: Changed.

**Revised manuscript**

[revised manuscript text omitted]